# THE EFFECTS OF OVERPARAMETERIZATION ON SHARPNESS-AWARE MINIMIZATION: AN EMPIRICAL AND THEORETICAL ANALYSIS

## ABSTRACT

Training an overparameterized neural network can yield minimizers of the same level of training loss and yet different generalization capabilities. With evidence that indicates a correlation between sharpness of minima and their generalization errors, increasing efforts have been made to develop an optimization method to explicitly find flat minima as more generalizable solutions. This sharpness-aware minimization (SAM) strategy, however, has not been studied much yet as to how overparameterization can actually affect its behavior. In this work, we analyze SAM under varying degrees of overparameterization and present both empirical and theoretical results that suggest a critical influence of overparameterization on SAM. Specifically, we first use standard techniques in optimization to prove that SAM can achieve a linear convergence rate under overparameterization in a stochastic setting. We also show based on a stability analysis that the solutions found by SAM are indeed flatter and have more uniformly distributed Hessian moments compared to those of SGD. These results are corroborated with our experiments that reveal a consistent trend that the generalization improvement made by SAM continues to increase as the model becomes more overparameterized. We further present that sparsity can open up an avenue for effective overparameterization in practice.

## 1 INTRODUCTION

The success of deep learning in recent years can be attributed to large neural networks of growing size: the deeper and wider they become, it tends to produce state-of-the-art results for various applications (Kaplan et al., 2020; Dehghani et al., 2023). Why do such large, potentially overparameterized, neural networks work well?

While a complete understanding remains elusive, research suggests that overparameterization can help multiple aspects of the learning process including even generalization (Neyshabur et al., 2017; Du & Lee, 2018). In particular, overparameterized neural networks show convexity-like behavior during optimization, making all local minima globally optimal, and thereby, enabling one to rely on local optimization methods such as gradient descent (Kawaguchi, 2016; Ma et al., 2018; Sagun et al., 2018).

However, not all local minima are necessarily equal as it turns out; for example, converging to different minima can yield a large disparity of generalization capabilities, despite their same level of training loss reaching almost zero (Keskar et al., 2017). One plausible explanation for such an implicit phenomenon is that generalization is somewhat negatively correlated with sharpness of the loss landscape, *i.e.*, flat minima tend to generalize better than sharp ones (Chaudhari et al., 2017; Jiang et al., 2020). This calls for new ways to guide the optimization process to converge to flat minima, and various strategies have been suggested to this end (Izmailov et al., 2018; Foret et al., 2021; Orvieto et al., 2022). Indeed, it has been observed in many cases that minimizing sharpness improves generalization performance (Bahri et al., 2022; Chen et al., 2022b; Qu et al., 2022).

Nevertheless, the sharpness minimization scheme has not been studied much under overparameterization as of yet, despite their contemporary relevance. Hence, we analyze the effects of overparameterization on sharpness-aware minimization, or SAM (Foret et al., 2021), in this work. Precisely, we develop convergence properties of SAM under overparameterization and characterize the linear

stability of the minima attained. We conduct a wide range of experiments to measure the effects of overparameterization to various extents. We also look into how sparsification, a model compression technique with increasing significance in the recent scaling trend, can alleviate the computational and memory overhead introduced by overparameterization.

Our key findings are summarized as follows: (i) SAM achieves a linear rate of convergence under overparameterization; (ii) the solutions found by SAM are flatter in shape and have a more uniformly distributed sharpness spectrum compared to SGD; (iii) the effectiveness of SAM on improving generalization performance increases as per increasing number of parameters; (iv) sparsity can be an effective strategy to restore the benefit of overparameterization for SAM without downscaling.

## 2 BACKGROUND

Consider the general unconstrained optimization problem:

$$\min_x f(x) \tag{1}$$

where $f : \mathbb{R}^d \to \mathbb{R}$ is the objective function to minimize, and $x \in \mathbb{R}^d$ is the optimization variable. In the context of deep learning, $f$ is usually an empirical risk and nonconvex, and $x$ is the parameters of a neural network which is often overparameterized. The standard approach to (1) is to run an iterative first-order method until it converges to a stationary point $x^\star$. Precisely, it has the following generalized form

$$x_{t+1} = x_t - \eta_t G(x_t; \xi_t) \tag{2}$$

where $\eta_t$ denotes a step size, and $G$ denotes a stochastic gradient estimate at the current iterate $x_t$ with $\xi_t$ referring to randomness. It is well known that for smooth and strongly convex functions GD converges linearly, whereas stochastic gradient descent (SGD) can only converge sublinearly (Ghadimi & Lan, 2013; Karimi et al., 2016; Bottou et al., 2018).

### 2.1 SHARPNESS-AWARE MINIMIZATION

Based on recent observations that indicate a correlation between the sharpness of $f$ at a minimum and its generalization error (Keskar et al., 2017; Jiang et al., 2020), Foret et al. (2021) suggest to turn (1) into a min-max problem of the following form

$$\min_x \max_{\|\epsilon\|_2 \leq \rho} f(x + \epsilon) \tag{3}$$

where $\epsilon$ and $\rho$ denote some perturbation added to $x$ and its bound, respectively; thus, the goal is now to seek $x$ that minimizes $f$ in its entire $\epsilon$-neighborhood, such that the objective landscape becomes flat. Taking the first-order Talyor approximation of $f$ at $x$ and solving for optimal $\epsilon^\star$ gives the following update rule for SAM:

$$x_{t+1} = x_t - \eta \nabla f \left( x_t + \rho \frac{\nabla f(x_t)}{\|\nabla f(x_t)\|_2} \right). \tag{4}$$

There is considerable evidence, albeit still debatable, that supports the effectiveness of SAM and its variants on improving generalization performance (Chen et al., 2022b; Kaddour et al., 2022; Bahri et al., 2022). Recently, it is proved that a stochastic version of SAM converges at a sublinear rate as with SGD (Andriushchenko & Flammarion, 2022).

### 2.2 OVERPARAMETERIZATION OF NEURAL NETWORKS

Here we introduce two concepts that are important for analyzing overparameterized neural networks.

Firstly, a neural network can be called overparameterized if it has a sufficient number of parameters to interpolate the whole training data, *i.e.*, it achieves zero training loss. From a stochastic optimization perspective, this means that a minimizer for the empirical risk $f$ also minimizes the risk for individual data point $f_i$. We formalize these observations as follows:

**Definition 1.** *(Interpolation) If $f(x^\star) = 0$ and $\nabla f(x^\star) = 0$, then $f_i(x^\star) = 0$ and $\nabla f_i(x^\star) = 0$ for $i = 1, \ldots, n$, where $n$ is the number of training data points.*

Next, a minimizer $x^\star$ is called linearly stable once arrived at a fixed point if it does not deviate far from $x^\star$. We formally define the linear stability as follows:

**Definition 2.** *(Linear stability) A minimizer $x^\star$ is linearly stable if there exists a constant $C$ such that $\mathbb{E}[\|\tilde{x}_t - x^\star\|^2] \leq C\|\tilde{x}_0 - x^\star\|^2$ for all $t > 0$ under $\tilde{x}_{t+1} = \tilde{x}_t - \nabla G(x^\star)(\tilde{x}_t - x^\star)$.*

Several work have been put forward in the literature to analyze convergence properties of SGD under different overparameterized regimes, including most notably Ma et al. (2018); Bassily et al. (2018), which have proved that SGD can converge much faster at a linear rate. Also, it has been shown that SGD can converge to flatter minima than GD under a linear stability analysis (Wu et al., 2022).

## 3 CONVERGENCE ANALYSIS OF SAM UNDER OVERPARAMETERIZATION

In this section, we show that a stochastic SAM converges linearly for an overparameterized model. We first provide a formal definition of relevant assumptions:

**Definition 3.** *(Smoothness) $f$ is $\beta$-smooth if there exists $\beta > 0$ s.t. $\|\nabla f(x) - \nabla f(y)\| \leq \beta\|x - y\|$ for all $x, y \in \mathbb{R}^d$.*

**Definition 4.** *(Polyak-Lojasiewicz) $f$ is $\alpha$-PL if there exists $\alpha > 0$ s.t. $\|\nabla f(x)\|^2 \geq \alpha(f(x) - f(x^\star))$ for all $x \in \mathbb{R}^d$.*

The smoothness assumption is standard in optimization literature and any neural network with smooth activation and loss function satisfies this. Also, the Polyak-Lojasiewicz (PL) condition is satisfied when the model is overparameterized (Belkin, 2021; Liu et al., 2022).

Next, we provide the following lemmas that are integral to proving the main theorem.

**Lemma 5.** *Suppose that $f_i$ is $\beta$-smooth. Then*

$$\langle \nabla f_i(x + \rho \nabla f_i(x)), \nabla f(x) \rangle \geq \langle \nabla f_i(x), \nabla f(x) \rangle - \frac{\beta\rho}{2}\|\nabla f_i(x)\|^2 - \frac{\beta\rho}{2}\|\nabla f(x)\|^2.$$

**Lemma 6.** *Suppose that $f_i$ is $\beta$-smooth. Then*

$$\|\nabla f_i(x_t + \rho \nabla f_i(x_t))\|^2 \leq (\beta\rho + 1)^2 \|\nabla f_i(x_t)\|^2.$$

These two lemmas essentially show that the stochastic SAM gradient, *i.e.*, $\nabla f_i(x + \rho \nabla f_i(x))$, aligns well with and scales to the standard stochastic gradient, *i.e.*, how similar SAM is to SGD. In fact, these results become tight as $\beta$ and $\rho$ decrease, which makes SAM reduce to SGD and matches our intuition. With these lemmas, we are ready to give the convergence result in the following theorem.

**Theorem 7.** *Suppose that $f_i$ is $\beta$-smooth, $f$ is $\lambda$-smooth and $\alpha$-PL, and interpolation holds. For any $\rho \leq \frac{1}{(\beta/\alpha + 1/2)\beta}$, a stochastic SAM that runs for $t$ iterations with step size $\eta^\star \overset{def}{=} \frac{\alpha - (\beta + \alpha/2)\beta\rho}{2\lambda\beta(\beta\rho + 1)^2}$ gives the following convergence guarantee:*

$$\mathbb{E}_{x_t}[f(x_t)] \leq \left(1 - \frac{\alpha - (\beta + \alpha/2)\beta\rho}{2}\eta^\star\right)^t f(x_0).$$

This result shows that a stochastic SAM converges at a linear rate under overparameterization. We provide the full proof for this theorem in Appendix D. We leave several remarks:

- When the perturbation bound is set to be zero, *i.e.*, $\rho = 0$, SAM reduces to SGD, and we obtain the well-known convergence rate for SGD in interpolated regime from Bassily et al. (2018).

- We provide in the full proof a convergence result for the more general case of a mini-batch SAM, from which a stochastic SAM is derived simply as a special case with the batch size of 1.

- The proof does not require the bounded gradient assumption, since the interpolation provides necessary guarantees. This suggests that overparameterization can ease the convergence of SAM.

- While the convergence of SAM has been studied in the literature under different settings, no previous work has developed a linear convergence rate of a stochastic SAM in an overparameterized regime.

To corroborate our result, we conduct experiments on three different empirical risk minimization tasks, *i.e.*, matrix factorization, MNIST classification, and CIFAR10 classification, and measure how training proceeds under varying parameterizations. The results are reported in Figure 1.

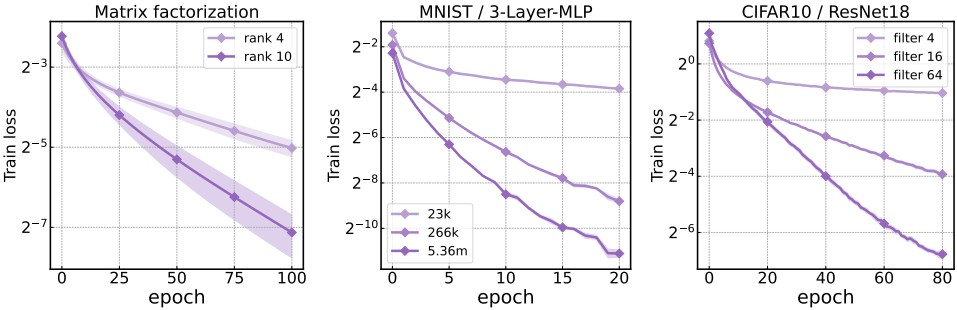

Figure 1: Empirical evaluation of convergence properties for SAM. When model is overparameterized, SAM converges much faster and closely to a linear rate. The convergence and training losses are consistent over several seeds, and by extension, consistent over minimizers. See Appendix A for the experiment details.

## 4 LINEAR STABILITY ANALYSIS OF SAM UNDER OVERPARAMETERIZATION

It has been observed in Woodworth et al. (2020); Xie et al. (2021) that SGD converges to certain types of minima among many others in an overparameterized regime. This raises the following question: which minima does SAM select and how do they differ from the minima found by SGD? We address this question from the perspective of linear stability (Wu et al., 2018; 2022) in this section.

We first define a linearized stochastic SAM, which is derived from applying first-order Taylor approximation on a stochastic SAM update given as follows:

**Definition 8.** *(Linearized stochastic SAM) We define a linearized stochastic SAM as*

$$\tilde{x}_{t+1} = \tilde{x}_t - \eta H_{\xi_t}(\tilde{x}_{t+1/2} - x^\star), \tag{5}$$

*where $\tilde{x}_{t+1/2} = \tilde{x}_t + \rho H_{\xi_t}(\tilde{x}_t - x^\star)$ is the linearized ascent step and $H_{\xi_t}$ is the Hessian estimation at step $t$.*

This actually corresponds to using SAM for the quadratic approximation of $f$ near $x^\star$, and we use this fact in the experiment setup. With this definition, we provide the following theorem for a linearly stable minima for a stochastic SAM.

**Theorem 9.** *Let us assume $x^\star = 0$ without loss of generality. If the following condition holds, the global minimum $x^\star$ is linearly stable for an unnormalized stochastic SAM.*

$$\begin{aligned}\lambda_{\max}\big((I - \eta H - \eta\rho H^2)^2 \\ + \eta(\eta - 2\rho)(M_2 - H^2) + 2\eta^2\rho(M_3 - H^3) + \eta^2\rho^2(M_4 - H^4)\big) \le 1\end{aligned} \tag{6}$$

*where $H = \frac{1}{n}\sum_{i=1}^n H_i$ and $M_k = \frac{1}{n}\sum_{i=1}^n H_i^k$ are the average Hessian and the $k$-th moment of the Hessian at $x^\star$.*

The detailed proof of the theorem is provided in Appendix E.

The above theorem tells us that the minima found by SAM will have an upper bound on the maximum eigenvalue of the Hessian related terms. To provide a more intuitive version of the theorem, we develop a necessary condition of (6). We obtain this by bounding each term in (6) to satisfy the stability condition:

$$0 \le a(1 + \rho a) \le \frac{2}{\eta}, \quad 0 \le s_2^2 \le \frac{1}{\eta(\eta - 2\rho)}, \quad 0 \le s_3^3 \le \frac{1}{2\eta^2\rho}, \quad 0 \le s_4^4 \le \frac{1}{\eta^2\rho^2}, \tag{7}$$

where $a = \lambda_{\max}(H)$, $s_k = \lambda_{\max}((M_k - H^k)^{1/k})$ are the sharpness and the non-uniformity of the Hessian measured with the $k$-th moment, respectively.

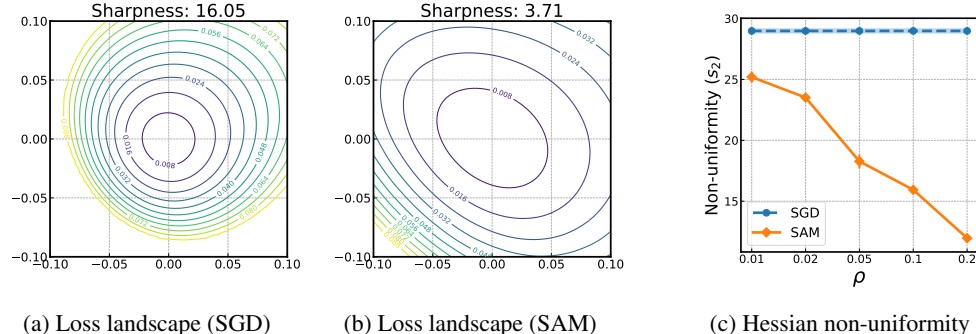

Figure 2: (a, b) Loss landscape along the direction of dominant eigenvectors and the corresponding sharpness $a = \lambda_{max}(H)$ between SAM and SGD. SAM converges to flatter minima with lower sharpness. (c) Non-uniformity of Hessian ($s_2$) between SAM and SGD. SAM indeed converges to minima with a smaller value of $s_2$.

The condition on $a$ indicates that SAM selects minima with bounded sharpness, in which the sharpness decreases as $\rho$ increases. Comparing with the necessary condition for SGD in Wu et al. (2018), *i.e.*, $0 \le a \le 2/\eta$, this result indicates that SAM selects flatter minima than SGD in the overparameterized regime. Next, the conditions on $s_2, s_3, s_4$ upper-bound the non-uniformity of Hessian at the minima found by SAM. We find that SAM puts additional constraints on $s_3$ and $s_4$ whereas SGD only upper bounds $s_2$, which may result in more uniform Hessian of the minima found by SAM compared to the case of SGD. We also remark that a larger $\rho$ makes the bounds on $s_3$ and $s_4$ smaller, which may lead to a more uniform Hessian spectrum.

To corroborate our result, we conduct experiments to evaluate empirical sharpness and non-uniformity of Hessian. Specifically, we set up an overparameterized MLP network for MNIST with squared loss, so that the local quadratic approximation becomes precise. We use $1,000$ random samples to calculate the non-uniformity, and all models are trained to reach near zero loss. The results are reported in Figure 2. We observe that SAM converges to flatter minima with lower sharpness compared to SGD. Also, the minima of SAM have a more uniformly distributed Hessian spectrum, which continues to decrease as $\rho$ increases. All these results align quite well with the necessary conditions (7).

## 5 EMPIRICAL EVALUATIONS OF SAM UNDER OVERPARAMETERIZATION

In this section, we evaluate the effect of overparameterization on SAM for various workloads of classification tasks, and empirically verify that the improvement in generalization performance by SAM over SGD consistently increases as with an increasing degree of overparameterization.

### 5.1 EXPERIMENT SETUP

We experiment with four different workloads: 3-layer MLP for MNIST, ResNet18 and Vision Transformers (ViTs) for CIFAR10, and ResNet50 for ImageNet. We scale the number of neurons, convolutional filters, or the dimension of hidden states to overparameterize MLP, ResNet, and ViTs respectively. We report results of the average over three random seeds and perform a hyperparameter search to select the optimal perturbation bound $\rho^\star$ (see Appendix A for the full details).

### 5.2 RESULTS

We first evaluate the effect of overparameterization on the generalization improvement of SAM, *i.e.*, the gap of validation accuracy between SAM and SGD. Interestingly, we find that the generalization benefit of SAM increases with an increasing number of parameters across all settings; in other words, SAM outperforms SGD with a larger margin in overparameterized regimes as seen in Figure 3. Specifically for Cifar-10 with ResNet18, the accuracy gap between SAM and SGD increases from around $0.3\%$ for networks with $45$ thousand parameters to $1.0\%$ for those with more than $11$ million parameters (see Section 5.2). We additionally observe that for this case the region with increased

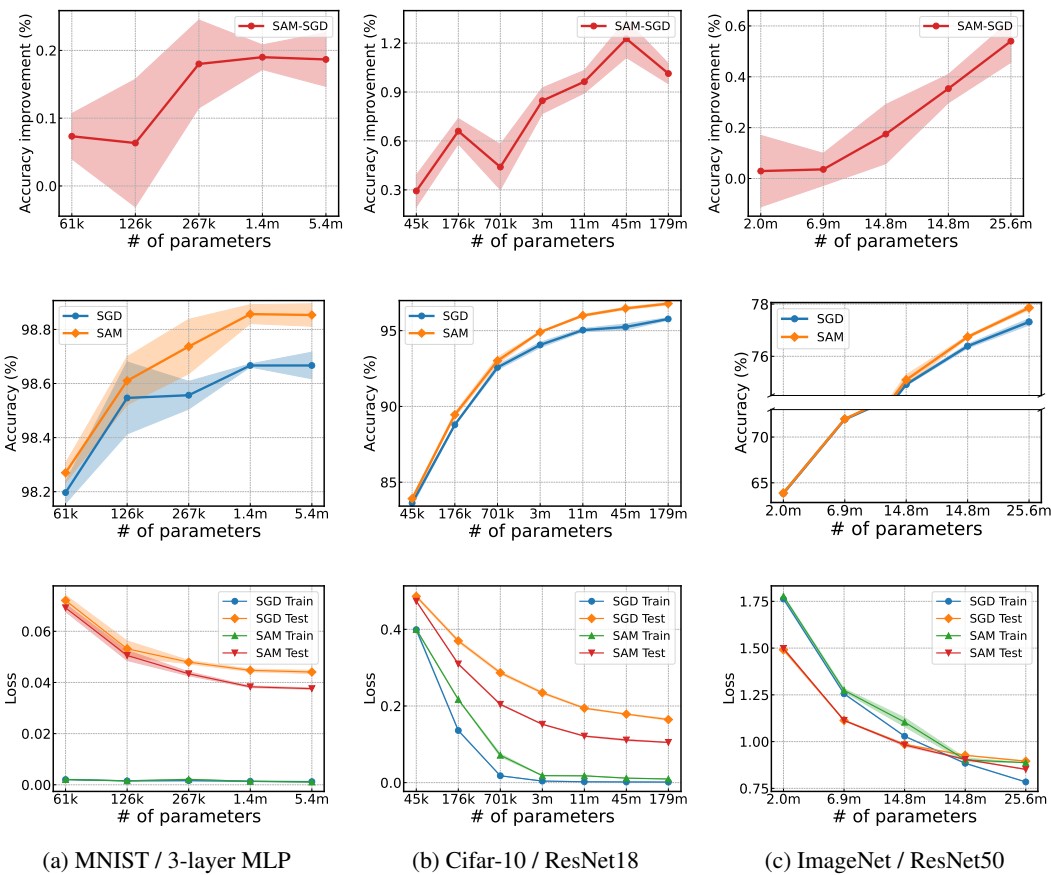

Figure 3: Effect of overparameterization on the generalization benefit of SAM for different settings. The top row represents the difference in validation accuracy between SAM and SGD while the middle and bottom rows represent the corresponding accuracy and loss values. The gap of validation accuracy between SAM and SGD tends to increase as with increasing number of parameters.

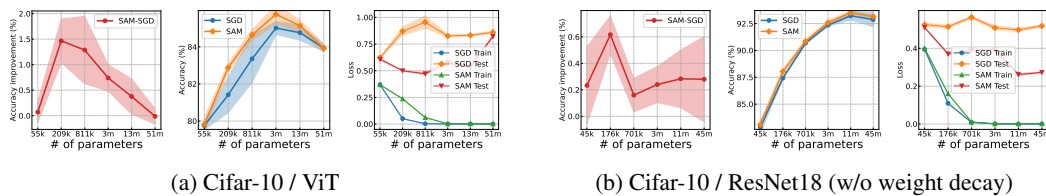

Figure 4: Effect of overparameterization on SAM for the ViT (a) and ResNet without weight decay (b) for Cifar-10. For each setting, we present the difference in validation accuracy, the corresponding accuracy values, and the loss values. The generalization benefit of SAM does not always increase with an increasing number of parameters in these cases where the networks can be easily overfitted from overparameterization.

generalization benefit of SAM has approximately zero training loss (see Figure 3b). We remark that the increased benefit of SAM is not attributed to linearization; it is due to the increased number of parameters as we discuss in more detail in Appendix C. Overall, the increased generalization performance of SAM with more parameters renders a promising avenue since modern neural network models are often heavily overparameterized (Zhang et al., 2022; Dehghani et al., 2023).

While the generalization benefit of SAM tends to increase as with more parameters for the previous cases, we observe it is not always the case for vision transformers. Specifically, the gap between

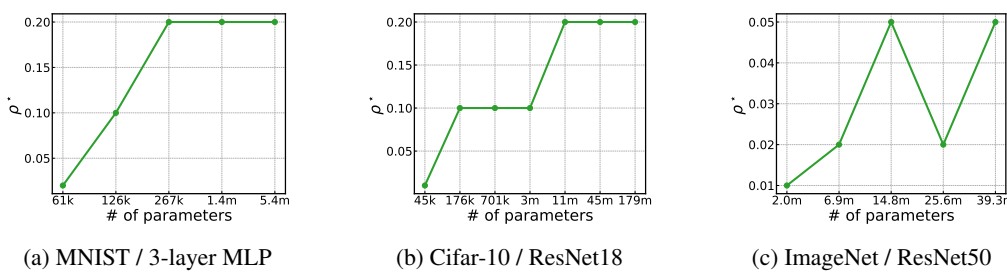

Figure 5: Optimal perturbation bound $\rho^\star$ for three workloads with varying number of parameters found by a hyperparameter search. $\rho^\star$ tends to increase as with increasing number of parameters.

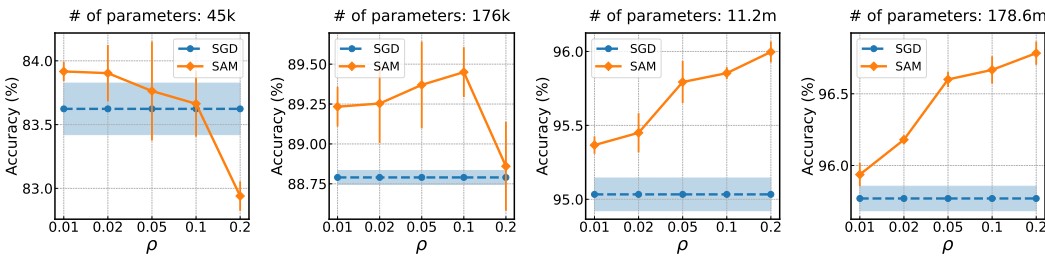

Figure 6: Validation accuracy versus $\rho$ for ResNet18 of different sizes trained on Cifar-10. All trends are rather continuous, validating the direct influence of $\rho$ on validation accuracy.

SAM and SGD gets smaller after exceeding 209k parameters (see Figure 4a). We suspect that this is because vision transformers are prone to overfitting small datasets without being pretrained on a massive dataset (Lee et al., 2021; Chen et al., 2022a), and thus, overparameterizing models without pretraining as in our setting may decrease the overall performance. Indeed, we find that the validation accuracy starts to decrease for both SAM and SGD after exceeding 3.2m parameters as seen in Figure 4a. Similar results of the non-increased generalization benefit of SAM with more parameters are also observed on ResNets without weight decay where the models can easily overfit, and validation accuracy drops after 11.2m parameters (see Figure 4b).

Moreover, we present the results of a hyperparameter search for optimal perturbation bound $\rho^\star$ in Figure 5. It is observed that $\rho^\star$ tends to increase as the model becomes more overparameterized; on Cifar-10 with ResNet18, for example, the smallest model has $\rho^\star = 0.01$ while the largest three have $\rho^\star = 0.2$. This result is quite intuitive in the following sense: the scale of perturbation $\epsilon$ controlled by $\rho$ needs to increase relatively proportionally with respect to the increase in model dimensionality, so as to preserve a similar level of perturbation effect; an extreme case is when the model grows infinitely large, $\rho$ should grow together, otherwise no perturbation can be made, *i.e.*, $\|\epsilon\| \to \infty$ as $d \to \infty$. We also plot the generalization performance corresponding to the search result in Figure 6, in which we find that the impact of different $\rho$ is directly reflected on validation performance as a continuous trend.

# 6 Sparse overparameterization for SAM

So far, we have observed how various aspects of SAM can be enhanced with overparameterization. While this result holds promise for the use of SAM given the recent trend of model scaling (Zhang et al., 2022; Dehghani et al., 2023), overparameterization can impose additional computational and memory burden during training and inference. In this section, we attempt to alleviate this problem via sparsification, a model compression technique that is widely used among others (Hoefler et al., 2021).

Specifically, we introduce a certain level of sparsity to an overparameterized model, such that the number of parameters matches to the original model; here, we try two sparsification methods that do not require pretraining, Random and SNIP (Lee et al., 2019). Then, we train several models of varying sparsity levels from scratch using SGD and SAM and compare their generalization performances.

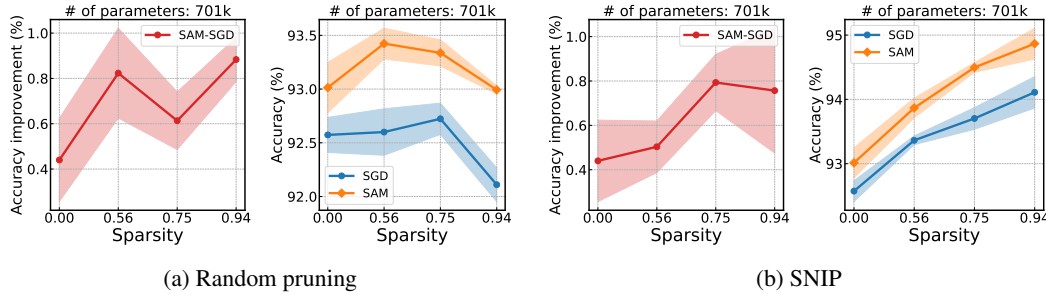

(a) Random pruning               (b) SNIP

Figure 7: Effect of sparsification on the generalization benefit of SAM (on Cifar-10 and ResNet18). The improvement tends to increase in large sparse models compared to their small dense counterparts. See the Appendix B for more results.

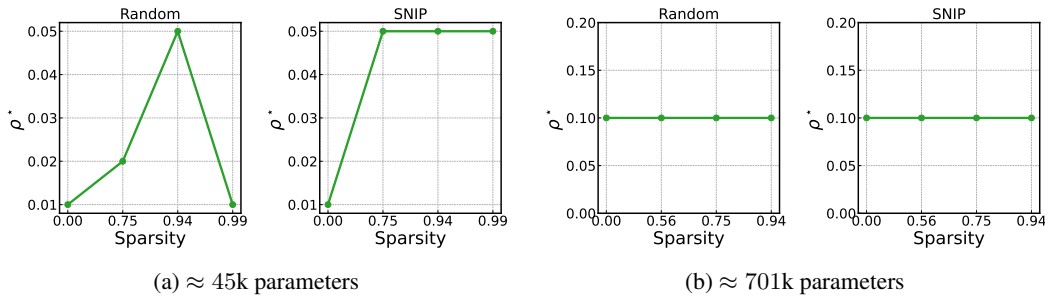

(a) $\approx$ 45k parameters             (b) $\approx$ 701k parameters

Figure 8: Effect of sparsification on $\rho^\star$ (on Cifar-10 and ResNet18). The sparsity pattern appears to be another factor to determine $\rho^\star$ alongside the parameter count. See Appendix B for more results.

As a result, we first observe that the generalization improvement of SAM from SGD tends to increase as the model becomes more sparsely overparameterized (see Figure 7). For example, with random pruning, the accuracy gap between SAM and SGD is around $0.4\%$ in small dense models, which increases to around $0.9\%$ in the overparameterized model at the extreme sparsity level of $94\%$; the trend seems more evident with SNIP which preserves the trainability of sparse models better than the naive random sparsity. While the fact that a large spare model outperforms the small dense counterpart has been observed before (Kalchbrenner et al., 2018; Lee et al., 2020), our result further suggests that one can consider taking sparsification more actively when using SAM.

Meanwhile, we find that $\rho^\star$ is sometimes different between small dense and large sparse models despite having a similar number of parameters; for the ResNet of $45$k parameters on Cifar-10, $\rho^\star$ changes over different sparsity levels and sparsification methods, but this does not generalize to the model of $701$k parameters (see Figure 8). This indicates that it is not just the parameter count that affects the behavior of SAM, but some other factors such as the pattern of parameterization.

## 7 RELATED WORK

Overparameterization advances the state of art in modern deep learning (Kaplan et al., 2020; Dehghani et al., 2023), benefits both optimization (Kawaguchi, 2016; Ma et al., 2018; Sagun et al., 2018) and generalization (Neyshabur et al., 2017; Du & Lee, 2018), and spurs many others toward understanding its fundamental nature. Belkin (2021); Liu et al. (2022) argue that overparameterized models possess Polyak-Lojasiewicz (PL) loss function (Lojasiewicz, 1963; Polyak, 1964). Ma et al. (2018); Bassily et al. (2018); Allen-Zhu et al. (2019) shows that it improves convergence rate from sublinear rate (Ghadimi & Lan, 2013; Karimi et al., 2016; Bottou et al., 2018) to linear rate by alleviating randomness around minima. For generalization, researchers point to implicit bias as the primary factor (Neyshabur, 2017; Zhang et al., 2021; Vardi, 2023). Wu et al. (2018; 2022) shows that SGD arrives at flat minima in this regime using linear stability. Cohen et al. (2021); Arora et al. (2022) draw similar conclusions for gradient descent with large learning rates, namely as edge of stability.

Many studies report a strong correlation between the flatness of minima and the generalization (Hochreiter & Schmidhuber, 1997; Keskar et al., 2017; Dziugaite & Roy, 2017; Neyshabur et al., 2017; Wei & Ma, 2019; Jiang et al., 2020), although there is a considerable debate (Dinh et al., 2017; Li et al., 2018; Andriushchenko et al., 2023). Inspired by this, Foret et al. (2021) proposes SAM to explicitly regularize sharpness and effectively improves generalization (Chen et al., 2022b; Kaddour et al., 2022; Bahri et al., 2022). Subsequently many work analyze various aspects of SAM and its variants such as convergence rate (Andriushchenko & Flammarion, 2022; Zhuang et al., 2022; Mi et al., 2022; Si & Yun, 2023), and implicit bias (Andriushchenko & Flammarion, 2022; Agarwala & Dauphin, 2023; Compagnoni et al., 2023; Wen et al., 2023).

## 8    DISCUSSION

In Section 3 we develop a linear convergence rate for a stochastic unnormalized version of SAM. This is partly to simplify analysis and based on the empirical findings in previous work that suggest no practical difference from the normalized version (Andriushchenko & Flammarion, 2022; Agarwala & Dauphin, 2023). Some recent work, however, reports that a normalization step can make a difference in theory (Si & Yun, 2023; Compagnoni et al., 2023). We also believe that it would be valuable to analyze various other sharpness minimization schemes (Izmailov et al., 2018; Orvieto et al., 2022).

In Section 4 we show that the minima attained using SAM are flatter and more stable compared to those of SGD. This result must not be confused with the generalization capability of SAM, and in fact, there is still a debate over whether sharpness can be considered as a generalization measure (Andriushchenko et al., 2023). While it may not suffice to uphold the benefit of SAM with respect to generalization, we are somewhat inclined to draw a positive link based on our consistent empirical findings throughout this work.

In Section 5 we present that the improvement by SAM over SGD increases as a model becomes more overparameterized. Considering the recent scaling trend (Zhang et al., 2022; Scao et al., 2022; Dehghani et al., 2023), it would be interesting to see whether our findings generalize to extremely overparameterized regimes. We note however that scaling to such degrees, while preserving the model architecture, is nontrivial due to limited computational resources. In addition, recent work reports that SAM can also be useful for finetuning pretrained models (Bahri et al., 2022; Chen et al., 2022b). We plan to extend our work to such scenarios.

In Section 6 we explore sparsification as a potential remedy for the overhead of overparameterization. While our results render some promises, there is a lot of room for improvement for it to be practical. To this end, we believe that some advanced compression techniques such as structured sparsity (Zhou et al., 2021) or even quantization (Gholami et al., 2022) can help realize the benefit of SAM in overparameterized settings.

## 9    CONCLUSION

In this work, we have analyzed the effects of overparameterization on several theoretical and empirical aspects of SAM including convergence properties, linear stability, and generalization improvement. We have also explored sparsification as a tool to compress overparameterized models and make them more practical. With exceptions, all our results indicate that SAM can indeed benefit from overparameterization, which is verified extensively through computational experiments. Nonetheless, there remain several limitations in our analysis, and we are planning to address them in future work.

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

# A  EXPERIMENTAL DETAILS

For all the experiments, we run the experiments with the same configurations over 3 different random seeds. For the experiments with SAM, we search the perturbation size $\rho$ over $[0.01, 0.02, 0.05, 0.1, 0.2]$.

**Matrix factorization**  For this experiment, we solve the following non-convex regression problem: $\min_{W_1,W_2} \mathbb{E}_{x\sim\mathcal{N}(0,I)} \|W_2W_1x - Ax\|^2$ which satisfies the PL-condition (Loizou et al., 2021). We choose $A \in \mathbb{R}^{10\times 6}$ and generate 1000 training samples, which are used for training a rank $k$ linear network with two matrix factors $W_1 \in \mathbb{R}^{k\times 6}$ and $W_2 \in \mathbb{R}^{10\times k}$. Here, interpolation is satisfied when rank $k = 10$. We train two linear networks $k \in \{4, 10\}$ for 100 epochs with a constant learning rate of 0.0005 and compare the convergence speed.

**MNIST / 3-layer MLP**  We train the models for 100 epochs with batch size 128. Learning rate is set to 0.1 initially and decays by 0.1 after 50% and 75% of the total epochs. We use momentum of 0.9 and weight decay of 0.0001. To investigate the effect of overparameterization on SAM, we scale the width of the hidden layers from $0.25\times$ to $10\times$ compared to LeNet-300-100 whil the relative proportions of the widths are preserved to $3 : 1$. For the experiments of Figure 2, we use the hidden layer size of $[3000, 1000]$ and train the networks with constant learning rate of 0.1 without weight decay or momentum. We also use squared loss for this setup.

**Cifar-10 / ResNet18**  We follow the similar procedure as in Andriushchenko & Flammarion (2022). We train the models for 200 epochs with batch size 128. Learning rate is set to 0.1 initially and decays by 0.1 after 50% and 75% of the total epochs. We use momentum of 0.9 and weight decay of 0.0005 unless stated otherwise. We use basic data augmentation of random horizontal flip and random cropping. To investigate the effect of overparameterization on SAM, we use the model width factor, *i.e.*, the number of convolutional filters in the first block, of $[4, 8, 16, 32, 64, 128, 256]$.

**Cifar-10 / ViT**  We use the similar setup as in Cifar-10 / ResNet18 case with a slight difference. We use cosine learning rate scheduling and the weight decay of 0.0001. We also fix the number of layers as 6 and use the patch size of $4 \times 4$, resulting in 64 patches for Cifar-10. To investigate the effect of overparameterization on SAM, we use the hidden dimension of $[32, 64, 128, 256, 512, 1024]$. We fix the dimension of MLP to be $2\times$ of the hidden dimension and number of attention heads to be $1/32\times$ of the hidden dimension.

**ImageNet / ResNet50**  We follow the similar procedure as in Du et al. (2022). Learning rate is set to 0.1 initially and the cosine scheduling is used with linear warmup of 5000 steps. We train the models for 90 epochs with batch size 512 with basic data augmentation. We use momentum of 0.9 and weight decay of 0.0001. To investigate the effect of overparameterization on SAM, we use the model width factor, *i.e.*, the number of convolutional filters in the first block, of $[16, 32, 48, 64, 80]$. We also experiment with $\rho = 0.0005$ for the models having 16 or 32 convolutional filters in the first block.

## B ADDITIONAL EXPERIMENTAL RESULTS

We plot the effect of overparameterization on the training and test loss of SAM and SGD in Figure 9, where we find that the difference between the test loss of SAM and SGD gets larger with increasing parameters. We also present the loss plots for Figure 4 in Figure 10. We plot the test accuracy of SAM over different values of $\rho$ for different setups in Figures 11 to 13. Additional results on the effect of sparsification on generalization benefit of SAM is plotted in Figures 14 and 15 and the effect on $\rho^\star$ in Figure 16 respectively.

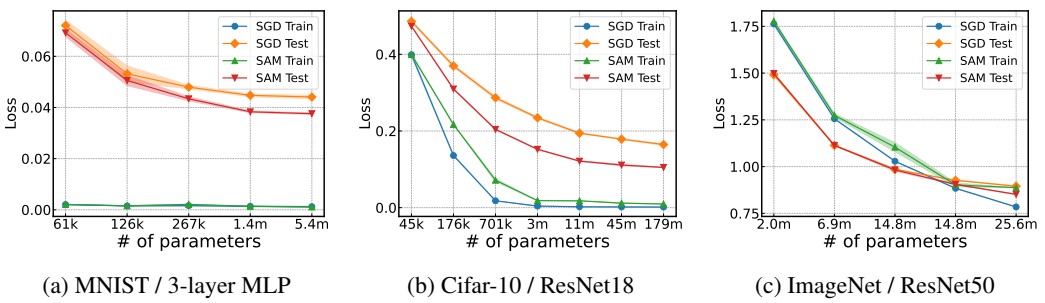

| (a) MNIST / 3-layer MLP | (b) Cifar-10 / ResNet18 | (c) ImageNet / ResNet50 |

Figure 9: Effect of overparameterization on training and test loss of SAM and SGD.

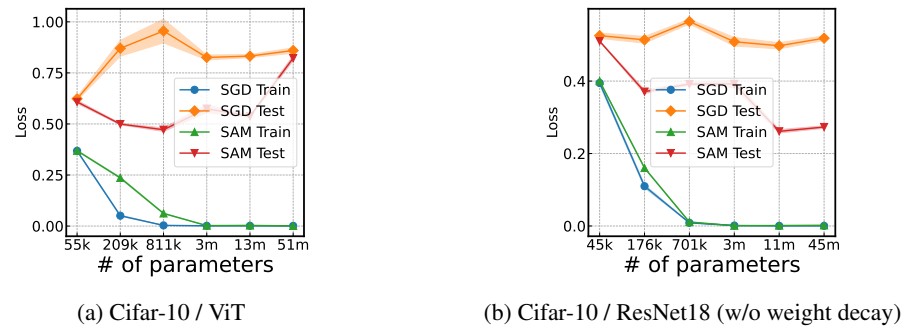

| (a) Cifar-10 / ViT | (b) Cifar-10 / ResNet18 (w/o weight decay) |

Figure 10: Effect of overparameterization on training and test loss of SAM and SGD for the ViT (a) and ResNet without weight decay (b).

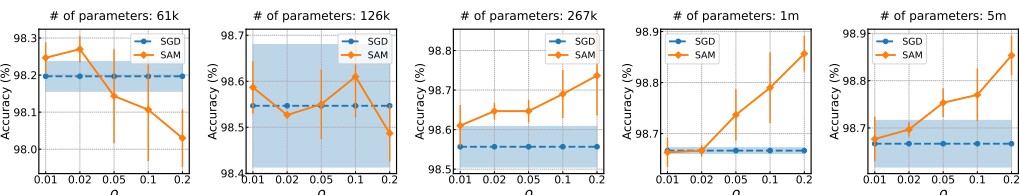

Figure 11: Test accuracy of SAM over different values of $\rho$ (for MNIST and 3-layer MLP).

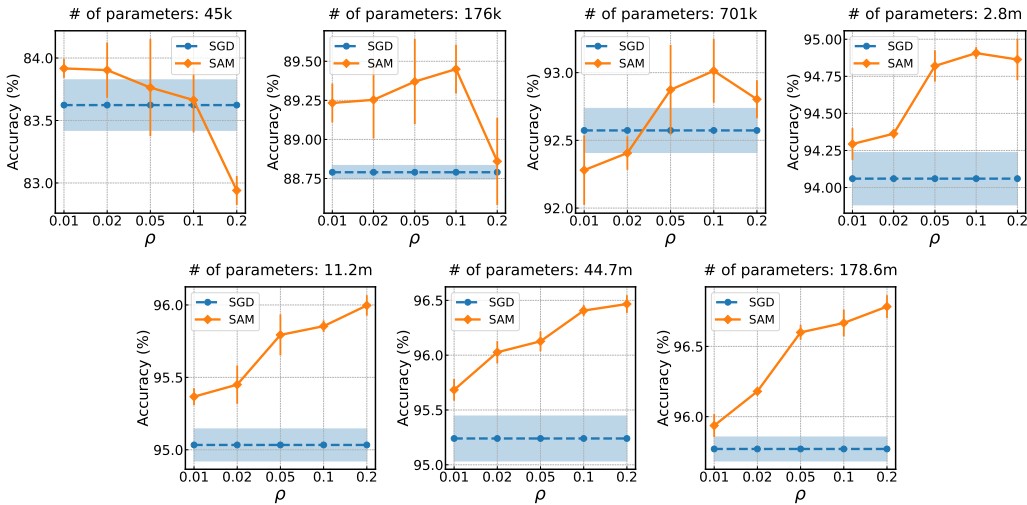

Figure 12: Test accuracy of SAM over different values of $\rho$ (for Cifar-10 and ResNet18).

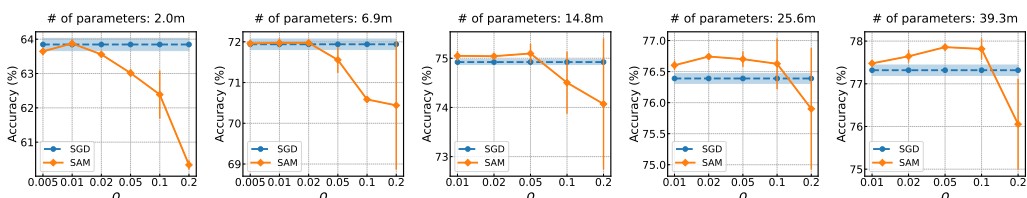

Figure 13: Test accuracy of SAM over different values of $\rho$ (for ImageNet and ResNet50).

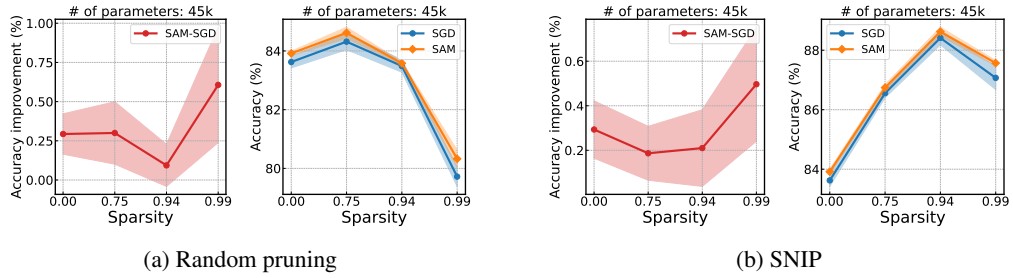

Figure 14: Effect of sparsification on generalization benefit of SAM on Cifar-10 and ResNet18 with 45k parameters.

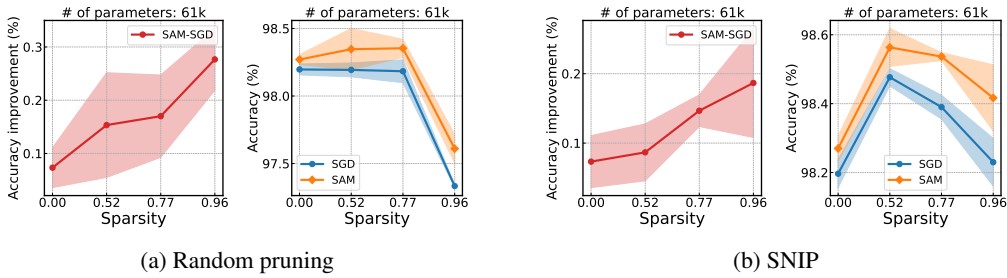

Figure 15: Effect of sparsification on generalization benefit of SAM on MNIST and 3-layer MLP with 61k parameters.

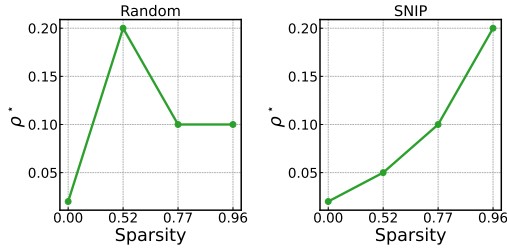

Figure 16: Effect of sparsification on $\rho^\star$. $\rho^\star$ can be sometimes different across different sparsity patterns despite having a similar number of parameters. We here plot the results with MNIST and 3-layer MLP.

## C   EFFECT OF SAM ON LINEARIZED REGIMES

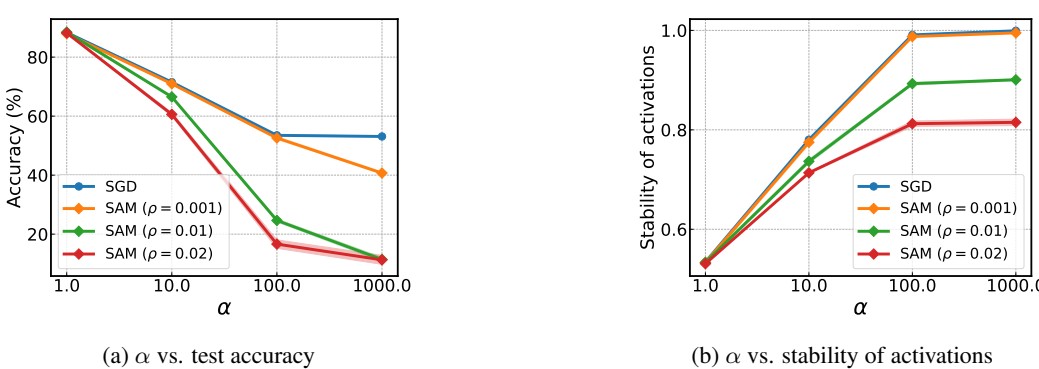

(a) $\alpha$ vs. test accuracy

(b) $\alpha$ vs. stability of activations

Figure 17: Effect of linearization on the performance of SGD and SAM.

It has been observed in the literature that highly overparameterized models can behave like linearized networks (Jacot et al., 2018) while the phenomenon not being specific to the overparameterized models (Chizat et al., 2019). One can thus wonder if the increased effectiveness of SAM observed in Section 5 directly comes from the overparameterization itself or is rather related to the linearization of the networks. In this section, we test how SAM works in the linearized regimes while fixing the number of parameters to be constant.

We follow the experiment setup of Chizat et al. (2019). Specifically, we train VGG-11 on the Cifar-10 dataset with the same hyperparameters including learning rate schedule, number of epochs, and batch size. We train the models with the $\alpha$-scaled squared loss $L(x, y) = \|f(x) - y/\alpha\|^2$ and use the centered model whose initial output is set to $0$. Here, a large value of $\alpha$ leads to a higher degree of linearization of the models. We report the stability of activations (see Appendix C.2 of Chizat et al. (2019) for how to compute), whose values close to $100\%$ imply an effective linearization.

We first observe that SAM underperforms SGD in the linearized regimes; while SGD and SAM with $\rho = 0.001$ both achieve effective linearization at $\alpha = 1000$, SAM underperforms SGD by more than $10\%$ in this case (see blue and orange curves in Figure 17). This suggests that the linearization does not improve the effect of SAM and implies that overparameterization itself is likely to be the cause of the improved performance of SAM.

We additionally observe the interesting point where SAM with a high value of $\rho$ does not achieve effective linearization by naively scaling $\alpha$ (see green and red curves in Figure 17b). We leave the thorough investigation of achieving effective linearization for SAM or analyzing the related phenomenon as future work.

# D    PROOF OF THEOREM 7

In this section, we show that a stochastic SAM converges linearly under an overparameterized regime. To put into perspective, this is the rate of convergence of gradient descent for a family of functions satisfying the PL-condition and smoothness assumptions (Karimi et al., 2016). We prove the convergence for an unnormalized mini-batch SAM given as

$$x_{t+1} = x_t - \eta g_t^B (x_t + \rho g_t^B (x_t)),$$

where $g_t^B(x) = \frac{1}{B} \sum_{i \in I_t^B} \nabla f_i(x)$ and $I_t^B \subseteq \{1, ..., n\}$ is a set of indices for data points in the mini-batch of size $B$ sampled at step $t$. This is a more general stochastic variant of SAM where a stochastic SAM in Section 3 is a particular case of a mini-batch SAM with mini-batch size $B = 1$.

We first make the following assumptions:

**(A1)** ($\beta$-smoothness of $f_i$). *There exists $\beta > 0$ s.t. $\|\nabla f_i(x) - \nabla f_i(y)\| \leq \beta \|x - y\|$ for all $x, y \in \mathbb{R}^d$,*

**(A2)** ($\lambda$-smoothness of $f$). *There exists $\lambda > 0$ such that $\|\nabla f(x) - \nabla f(y)\| \leq \lambda \|x - y\|$ for all $x, y \in \mathbb{R}^d$,*

**(A3)** ($\alpha$-PLness of $f$). *There exists $\alpha > 0$ s.t. $\|\nabla f(x)\|^2 \geq \alpha(f(x) - f(x^\star))$ for all $w, v \in \mathbb{R}^d$,*

**(A4)** (Interpolation). *If $f(x^\star) = 0$ and $\nabla f(x^\star) = 0$, then $f_i(x^\star) = 0$ and $\nabla f_i(x^\star) = 0$ for $i = 1, \ldots, n$, where $n$ is the number of training data points.*

Before we prove the main theorem, we first introduce two lemma important to the proof.

**Lemma 10.** *Suppose that Assumption (A1) holds. Then*

$$\langle \nabla f_i(x_{t+1/2}), \nabla f(x_t) \rangle \geq \langle \nabla f_i(x_t), \nabla f(x_t) \rangle - \frac{\beta \rho}{2} \|\nabla f_i(x_t)\|^2 - \frac{\beta \rho}{2} \|\nabla f(x_t)\|^2, \qquad (8)$$

*where $x_{t+1/2} = x_t + \rho \nabla f_i(x_t)$.*

This lemma shows how well a stochastic SAM gradient $\nabla f_i(x_{t+1/2})$ aligns with the true gradient $\nabla f(x_t)$. The two gradients become less aligned as $\beta$ and $\rho$ grow bigger, *i.e.* for sharper landscape and larger perturbation size.

*Proof.* We first add and subtract $\nabla f_i(x_t)$ on the left side of the inner product

$$\langle \nabla f_i(x_{t+1/2}), \nabla f(x_t) \rangle = \underbrace{\langle \nabla f_i(x_{t+1/2}) - \nabla f_i(x_t), \nabla f(x_t) \rangle}_{\tau_1} + \langle \nabla f_i(x_t), \nabla f(x_t) \rangle. \qquad (9)$$

We here bound the term $\tau_1$ so that it becomes an equality when $\rho = 0$. To achieve this, we start with the following binomial square, which is trivially lower bounded by 0.

$$0 \leq \frac{1}{2} \|\nabla f_i(x_{t+1/2}) - \nabla f_i(x_t) + \beta \rho \nabla f(x_t)\|^2$$

We then expand the above binomial square so that the term containing $\tau_1$ appears.

$$0 \leq \frac{1}{2} \|\nabla f_i(x_{t+1/2}) - \nabla f_i(x_t)\|^2 + \underbrace{\langle \nabla f_i(x_{t+1/2}) - \nabla f_i(x_t), \, \beta \rho \nabla f(x_t) \rangle}_{\beta \rho \tau_1} + \frac{1}{2} \|\beta \rho \nabla f(x_t)\|^2$$

We subtract the term $\beta \rho \tau_1$ on both sides of the inequality which gives

$$-\langle \nabla f_i(x_{t+1/2}) - \nabla f_i(x_t), \, \beta \rho \nabla f(x_t) \rangle \leq \frac{1}{2} \|\nabla f_i(x_{t+1/2}) - \nabla f_i(x_t)\|^2 + \frac{\beta^2 \rho^2}{2} \|\nabla f(x_t)\|^2.$$

Then we upper bound the right-hand side using the Assumption (A1):

$$-\langle \nabla f_i(x_{t+1/2}) - \nabla f_i(x_t) \,,\; \beta\rho\nabla f(x_t)\rangle \leq \frac{\beta^2}{2}\|x_{t+1/2} - x\|^2 + \frac{\beta^2\rho^2}{2}\|\nabla f(x_t)\|^2$$
$$= \frac{\beta^2\rho^2}{2}\|\nabla f_i(x_t)\|^2 + \frac{\beta^2\rho^2}{2}\|\nabla f(x_t)\|^2.$$

We divide both sides with $\beta\rho$, obtaining:

$$-\langle \nabla f_i(x_{t+1/2}) - \nabla f_i(x_t), \nabla f(x_t)\rangle \leq \frac{\beta\rho}{2}\|\nabla f_i(x_t)\|^2 + \frac{\beta\rho}{2}\|\nabla f(x_t)\|^2.$$

Applying this to (9) gives the bound in the lemma statement. $\qquad\square$

**Lemma 11.** *Suppose that Assumption (A1) holds. Then*

$$\left\|\nabla f_i(x_{t+1/2})\right\|^2 \leq (\beta\rho + 1)^2\|\nabla f_i(x_t)\|^2, \tag{10}$$

*where* $x_{t+1/2} = x_t + \rho\nabla f_i(x_t)$.

This second lemma shows that the norm of a stochastic SAM gradient is bounded by the norm of the stochastic gradient. Similar to the Lemma 10, as $\beta$ and $\rho$ grow bigger the norm for a stochastic SAM gradient can become larger than the norm of the true gradient.

*Proof.* We use the following binomial squares:

$$\|\nabla f_i(x_{t+1/2})\|^2$$
$$= \|\nabla f_i(x_{t+1/2}) - \nabla f_i(x_t)\|^2 + 2\langle\nabla f_i(x_{t+1/2}) - \nabla f_i(x_t), \nabla f_i(x_t)\rangle + \|\nabla f_i(x_t)\|^2.$$

We bound the right-hand side using Cauchy-Schwarz inequality and Assumption (A1), which gives

$$\left\|\nabla f_i(x_{t+1/2})\right\|^2$$
$$= \|\nabla f_i(x_{t+1/2}) - \nabla f_i(x_t)\|^2 + 2\langle\nabla f_i(x_{t+1/2}) - \nabla f_i(x_t), \nabla f_i(x_t)\rangle + \|\nabla f_i(x_t)\|^2$$
$$\underset{\text{C.S.}}{\leq} \|\nabla f_i(x_{t+1/2}) - \nabla f_i(x_t)\|^2 + 2\|\nabla f_i(x_{t+1/2}) - \nabla f_i(x_t)\|\|\nabla f_i(x_t)\| + \|\nabla f_i(x_t)\|^2$$
$$\underset{\text{(A1)}}{\leq} \beta^2\|x_{t+1/2} - x_t\|^2 + 2\beta\|x_{t+1/2} - x_t\|\|\nabla f_i(x_t)\| + \|\nabla f_i(x_t)\|^2$$
$$= \beta^2\rho^2\|\nabla f_i(x_t)\|^2 + 2\beta\rho\|\nabla f_i(x_t)\|^2 + \|\nabla f_i(x_t)\|^2$$
$$= (\beta\rho + 1)^2\|\nabla f_i(x_t)\|^2$$

$\qquad\square$

These two lemmas essentially show how similar a stochastic SAM gradient is to the stochastic gradient, where the two become more similar as $\beta$ and $\rho$ decrease, which aligns well with our intuition. Using Lemma 10 and 11, we provide the convergence result in the following theorem.

**Theorem 12.** *Suppose that Assumptions (A1-4) holds. For any mini-batch size $B \in \mathbb{N}$ and $\rho \leq \frac{1}{(\beta/\alpha+1/2)\beta}$, unnormalized mini-batch SAM with step size $\eta_B^\star \overset{def}{=} \frac{1-(\kappa_B+1/2)\beta\rho}{2\lambda\kappa_B(\beta\rho+1)^2}$ gives the following guarantee at step $t$:*

$$\mathbb{E}_{x_t}[f(x_t)] \leq \left(1 - \frac{\alpha\eta_B^\star}{2}\left(1 - \left(\kappa_B + \frac{1}{2}\right)\beta\rho\right)\right)^t f(x_0),$$

*where* $\kappa_B = \frac{1}{B}\left(\frac{B-1}{2} + \frac{\beta}{\alpha}\right)$.

This theorem states that mini-batch SAM converges at a linear rate under overparameterization.

*Proof.* Proof can be outlined in 3 steps.

> **step 1.** Handle terms containing mini-batch SAM gradient $g_t^B(x_t + \rho g_t^B(x_t))$ using bounds from **(A1)**.
>
> **step 2.** Take conditional expectation $\mathbb{E}[\cdot \,|\, x_t]$ and substitute expectation of function of mini-batch gradient $g_t^B$ with terms containing $\|\nabla f(x_t)\|$ and $\mathbb{E}\left[\|\nabla f_i(x_t)\|^2 \,\middle|\, x_t\right]$.
>
> **step 3.** Bound the two terms from **step 2**, one using Assumptions **(A2)** and **(A4)** and the other using Assumption **(A3)** which results in all the terms to contain $f(x_t)$. Then finally we take total expectations to derive the final runtime bound.

We start from the quadratic upper bound derived from Assumption **(A2)**;

$$f(x_{t+1}) \leq f(x_t) + \langle \nabla f(x_t),\, x_{t+1} - x_t \rangle + \frac{\lambda}{2}\|x_{t+1} - x_t\|^2.$$

Applying mini-batch SAM update, we then have

$$f(x_t) - f(x_{t+1}) \geq \eta \left\langle \nabla f(x_t),\, g_t^B(x_{t+1/2}) \right\rangle - \frac{\eta^2 \lambda}{2} \left\| g_t^B(x_{t+1/2}) \right\|^2,$$

where $x_{t+1/2} = x_t + \rho g_t^B(x_t)$.

**step 1.** We can see that there are two terms that contain a mini-batch SAM gradient $g_t^B(x_{t+1/2})$. We see that each can be bounded directly using Lemma 10 and 11, which gives

$$
\begin{aligned}
f(x_t) - f(x_{t+1}) &\geq \eta \left( \langle g_t^B(x_t), \nabla f(x_t) \rangle - \frac{\beta \rho}{2}\|g_t^B(x_t)\|^2 - \frac{\beta \rho}{2}\|\nabla f(x_t)\|^2 \right) \\
&\quad - \frac{\eta^2 \lambda}{2}(\beta \rho + 1)^2 \|g_t^B(x_t)\|^2 \\
&= \eta \langle g_t^B(x_t), \nabla f(x_t) \rangle - \frac{\eta \beta \rho}{2}\|\nabla f(x_t)\|^2 - \frac{\eta}{2}\left(\eta \lambda (\beta \rho + 1)^2 + \beta \rho\right)\|g_t^B(x_t)\|^2.
\end{aligned}
$$

**step 2.** Now we apply $\mathbb{E}[\cdot \,|\, x_t]$ to all the terms.

$$
\begin{aligned}
\mathbb{E}\left[f(x_t) - f(x_{t+1}) \,\middle|\, x_t\right] &= f(x_t) - \mathbb{E}\left[f(x_{t+1}) \,\middle|\, x_t\right] \\
&\geq \eta \mathbb{E}\left[\langle g_t^B(x_t), \nabla f(x_t) \rangle \,\middle|\, x_t\right] - \frac{\eta \beta \rho}{2}\mathbb{E}\left[\|\nabla f(x_t)\|^2 \,\middle|\, x_t\right] \\
&\quad - \frac{\eta}{2}\left(\eta \lambda (\beta \rho + 1)^2 + \beta \rho\right)\mathbb{E}\left[\|g_t^B(x_t)\|^2 \,\middle|\, x_t\right] \\
&= \eta \left(1 - \frac{\beta \rho}{2}\right)\|\nabla f(x_t)\|^2 - \frac{\eta}{2}\left(\eta \lambda (\beta \rho + 1)^2 + \beta \rho\right)\mathbb{E}\left[\|g_t^B(x_t)\|^2 \,\middle|\, x_t\right].
\end{aligned}
$$

Here we expand the term $\mathbb{E}\left[\|g_t^B(x_t)\|^2 \,\middle|\, x_t\right]$ by expanding the mini-batched function into individual function estimators as follows.

$$\mathbb{E}_{g_t^B}\left[\left\|g_t^B(x_t)\right\|^2 \;\Big|\; x_t\right]$$

$$= \mathbb{E}_{I_t^B}\left[\left\langle \frac{1}{B}\sum_{i\in I_t^B}\nabla f_i(x_t)\,,\; \frac{1}{B}\sum_{j\in I_t^B}\nabla f_j(x_t)\right\rangle \;\Big|\; x_t\right]$$

$$= \frac{1}{B^2}\left\{\sum_{i\in I_t^B}\mathbb{E}_{f_i}\left[\|\nabla f_i(x_t)\|^2 \;\Big|\; x_t\right] + \sum_{i\in I_t^B}\sum_{\substack{j\in I_t^B \\ (j\neq i)}}\mathbb{E}_{f_i,f_j}\left[\langle\nabla f_i(x_t),\nabla f_j(x_t)\rangle \;\Big|\; x_t\right]\right\} \tag{11}$$

$$= \frac{1}{B}\mathbb{E}\left[\|\nabla f_i(x_t)\|^2 \;\Big|\; x_t\right] + \frac{B-1}{B}\|\nabla f(x_t)\|^2.$$

Using (11), we get

$$f(x_t) - \mathbb{E}\left[f(x_{t+1})\mid x_t\right] \geq \eta\left(1-\frac{\beta\rho}{2}\right)\|\nabla f(x_t)\|^2$$

$$-\frac{\eta}{2}\left(\eta\lambda\,(\beta\rho+1)^2+\beta\rho\right)\left(\frac{1}{B}\mathbb{E}\left[\|\nabla f_i(x_t)\|^2 \;\Big|\; x_t\right] + \frac{B-1}{B}\|\nabla f(x_t)\|^2\right)$$

$$= \eta\left(\left(1-\frac{\beta\rho}{2}\right) - \frac{B-1}{2B}\left(\eta\lambda(\beta\rho+1)^2+\beta\rho\right)\right)\|\nabla f(x_t)\|^2$$

$$-\frac{\eta}{2B}\left(\eta\lambda(\beta\rho+1)^2+\beta\rho\right)\mathbb{E}\left[\|\nabla f_i(x_t)\|^2 \;\Big|\; x_t\right]. \tag{12}$$

$\boxed{\textbf{step 3.}}$ In this step, we bound the two terms and take the total expectation to derive the final runtime bound.

We first derive a bound for $\mathbb{E}\left[\|\nabla f_i(x_t)\|^2 \;\Big|\; x_t\right]$. We start from the following bound derived from Assumption **(A1)**:

$$\|\nabla f_i(x_t) - \nabla f_i(x^\star)\|^2 \leq 2\beta(f_i(x_t) - f_i(x^\star)).$$

By Assumption **(A4)**, this reduces to

$$\|\nabla f_i(x_t)\|^2 \leq 2\beta f_i(x_t).$$

Applying this to (12) gives

$$f(x_t) - \mathbb{E}\left[f(x_{t+1})\mid x_t\right] \geq \eta\left(\left(1-\frac{\beta\rho}{2}\right) - \frac{B-1}{2B}\left(\eta\lambda(\beta\rho+1)^2+\beta\rho\right)\right)\|\nabla f(x_t)\|^2$$

$$-\frac{\eta\beta}{B}\left(\eta\lambda(\beta\rho+1)^2+\beta\rho\right)\mathbb{E}[f_i(x_t)|x_t]$$

$$= \eta\underbrace{\left(\left(1-\frac{\beta\rho}{2}\right) - \frac{B-1}{2B}\left(\eta\lambda(\beta\rho+1)^2+\beta\rho\right)\right)}_{\tau_2}\|\nabla f(x_t)\|^2$$

$$-\frac{\eta\beta}{B}\left(\eta\lambda(\beta\rho+1)^2+\beta\rho\right)f(x_t). \tag{13}$$

Next, to bound $\|\nabla f(x_t)\|^2$, we use the following bound derived from applying $f(x^*)=0$ to **(A3)**:

$$\|\nabla f(x)\|^2 \geq \alpha f(x). \tag{14}$$

Assuming $\tau_2 \geq 0$ which we provide a sufficient condition at the end of the proof, we apply (14) to (13) which gives

$$
\begin{aligned}
f(x_t) &- \mathbb{E}\big[f(x_{t+1}) \mid x_t\big] \\
&\geq \eta\alpha\left(\left(1 - \frac{\beta\rho}{2}\right) - \frac{B-1}{2B}\left(\eta\lambda(\beta\rho+1)^2 + \beta\rho\right)\right) f(x_t) - \frac{\eta\beta}{B}\left(\eta\lambda(\beta\rho+1)^2 + \beta\rho\right) f(x_t) \\
&= \eta\left(\alpha - \alpha\Big(\underbrace{\frac{1}{B}\Big(\frac{B-1}{2} + \frac{\beta}{\alpha}\Big)}_{\kappa_B} + \frac{1}{2}\Big)\beta\rho - \eta(\beta\rho+1)^2 \underbrace{\frac{\lambda}{B}\Big(\alpha\frac{B-1}{2} + \beta\Big)}_{\lambda\alpha\kappa_B}\right) f(x_t) \\
&= \eta\alpha\left(1 - \left(\kappa_B + \frac{1}{2}\right)\beta\rho - \eta\lambda(\beta\rho+1)^2\kappa_B\right) f(x_t).
\end{aligned}
$$

Hence, we get

$$
\mathbb{E}\big[f(x_{t+1}) \mid x_t\big] \leq \left(1 - \eta\alpha\Big(1 - \Big(\kappa_B + \frac{1}{2}\Big)\beta\rho\Big) + \eta^2\alpha\lambda(\beta\rho+1)^2\kappa_B\right) f(x_t).
$$

Applying total expectation on both sides gives

$$
\mathbb{E}[f(x_{t+1})] \leq \left(1 - \eta\alpha\Big(1 - \Big(\kappa_B + \frac{1}{2}\Big)\beta\rho\Big) + \eta^2\alpha\lambda(\beta\rho+1)^2\kappa_B\right) \mathbb{E}[f(x_t)]. \tag{15}
$$

Optimizing the multiplicative term in (15) with respect to $\eta$ gives $\eta = \frac{1-(\kappa_B+1/2)\beta\rho}{2\lambda\kappa_B(\beta\rho+1)^2}$, which is $\eta_B^\star$ in the theorem statement. With assumption of $\rho \leq \frac{1}{(\beta/\alpha+1/2)\beta}$ so that we have $\eta_B^\star \geq 0$, applying this to (15) gives

$$
\mathbb{E}\left[f(x_{t+1})\right] \leq \left(1 - \frac{\alpha\,\eta_B^\star}{2}\Big(1 - \Big(\kappa_B + \frac{1}{2}\Big)\beta\rho\Big)\right) \mathbb{E}\left[f(x_t)\right],
$$

which provides the desired convergence rate.

Last but not least, we calculate the upper bound for $\rho$ to satisfy the assumption $\tau_2 \geq 0$ by substituting $\eta$ for $\eta_B^\star$ in $\tau_2$, yielding $\rho \leq \frac{2B\kappa_B+2\beta/\alpha}{(2B-1)\kappa_B+\beta/\alpha}\frac{1}{\beta}$. Minimizing this upper bound with respect to $B$ gives $\rho \leq \frac{1}{\beta}$, which is a looser bound than $\rho \leq \frac{1}{(\beta/\alpha+1/2)\beta}$. $\qquad \square$

# E PROOF OF THEOREM 9

Here, we provide the detailed proof of Theorem 9.

Recall the linearized SAM from Equation (5). For simplicity, we denote $\tilde{x}_t$ as $x_t$, and we assume without loss of generality that the fixed point $x^\star = 0$. Then, the linearized SAM can be written as

$$
x_{t+1} = x_t - \eta H_{\xi_t}(x_t + \rho H_{\xi_t} x_t). \tag{16}
$$

Our goal is to derive a bound of the form $\mathbb{E}\|x_t\|^2 \leq C\|x_0\|^2$. We first apply (16) to $\mathbb{E}\big[\|x_{t+1}\|^2 \mid x_t\big]$ and continue expanding the terms as follows:

$$
\begin{aligned}
\mathbb{E}\left[\|x_{t+1}^2\| \mid x_t\right] &= \mathbb{E}\|x_t - \eta H_{\xi_t}(x_t + \rho H_{\xi_t} x_t)\|^2 \\
&= x_t^\top \mathbb{E}\left[\left(I - \eta H_{\xi_t} - \eta\rho H_{\xi_t}^2\right)^2 \mid x_t\right] x_t \\
&= x_t^\top \mathbb{E}\left[I - 2\eta(H_{\xi_t} + \rho H_{\xi_t}^2) + \eta^2\left(H_{\xi_t} + \rho H_{\xi_t}^2\right)^2 \mid x_t\right] x_t \\
&= x_t^\top \mathbb{E}\left[I - 2\eta(H_{\xi_t} + \rho H_{\xi_t}^2) + \eta^2\left(H_{\xi_t}^2 + 2\rho H_{\xi_t}^3 + \rho^2 H_{\xi_t}^4\right) \mid x_t\right] x_t \\
&= x_t^\top \mathbb{E}\left[I - 2\eta H_{\xi_t} + \eta(\eta - 2\rho)H_{\xi_t}^2 + 2\eta^2\rho H_{\xi_t}^3 + \eta^2\rho^2 H_{\xi_t}^4 \mid x_t\right] x_t \\
&= x_t^\top \left(I - 2\eta H + \eta(\eta - 2\rho)\mathbb{E}H_{\xi_t}^2 + 2\eta^2\rho\mathbb{E}H_{\xi_t}^3 + \eta^2\rho^2\mathbb{E}H_{\xi_t}^4\right) x_t \\
&= x_t^\top \Big(I - 2\eta H + \eta(\eta - 2\rho)H^2 + 2\eta^2\rho H^3 + \eta^2\rho^2 H^4 \\
&\qquad + \eta(\eta - 2\rho)(\mathbb{E}H_{\xi_t}^2 - H^2) + 2\eta^2\rho(\mathbb{E}H_{\xi_t}^3 - H^3) + \eta^2\rho^2(\mathbb{E}H_{\xi_t}^4 - H^4)\Big) x_t \\
&= x_t^\top \Big(\left(I - \eta H - \eta\rho H^2\right)^2 \\
&\qquad + \eta(\eta - 2\rho)(\mathbb{E}H_{\xi_t}^2 - H^2) + 2\eta^2\rho(\mathbb{E}H_{\xi_t}^3 - H^3) + \eta^2\rho^2(\mathbb{E}H_{\xi_t}^4 - H^4)\Big) x_t
\end{aligned}
$$

Since $x^\top A x \le \lambda_{\max}(A)\|x\|^2$ always holds for any $x$ and any matrix $A$ with the maximum eigenvalue $\lambda_{\max}(A)$, applying this inequality and taking the total expectation gives the following;

$$
\begin{aligned}
\mathbb{E}\left[\|x_{t+1}\|^2\right] \le \lambda_{\max}\Big( &\left(I - \eta H - \eta\rho H^2\right)^2 + \eta(\eta - 2\rho)(\mathbb{E}H_\xi^2 - H^2) \\
&+ 2\eta^2\rho(\mathbb{E}H_\xi^3 - H^3) + \eta^2\rho^2(\mathbb{E}H_\xi^4 - H^4)\Big)\mathbb{E}\left[\|x_t\|^2\right].
\end{aligned}
$$

Recursively applying this bound gives

$$
\begin{aligned}
\mathbb{E}\|x_t\|^2 \le \lambda_{\max}\Big( &\left(I - \eta H - \eta\rho H^2\right)^2 + \eta(\eta - 2\rho)(\mathbb{E}H_\xi^2 - H^2) \\
&+ 2\eta^2\rho(\mathbb{E}H_\xi^3 - H^3) + \eta^2\rho^2(\mathbb{E}H_\xi^4 - H^4)\Big)^t \|x_0\|^2.
\end{aligned}
$$

Here, we can see that $x^\star$ is linearly stable if

$$
\begin{aligned}
\lambda_{\max}\Big( &(I - \eta H - \eta\rho H^2)^2 \\
&+ \eta(\eta - 2\rho)(\mathbb{E}H_\xi^2 - H^2) + 2\eta^2\rho(\mathbb{E}H_\xi^3 - H^3) + \eta^2\rho^2(\mathbb{E}H_\xi^4 - H^4)\Big) \le 1.
\end{aligned}
$$

$\qquad\qquad\qquad\qquad\qquad\qquad\qquad\qquad\qquad\qquad\qquad\qquad\qquad\qquad\qquad\qquad\quad \square$

