# OpenReview forum: "The Effects of Overparameterization on Sharpness-aware Minimization: An Empirical and Theoretical Analysis"
_ICLR.cc/2024/Conference — Submitted to ICLR 2024_

### Official Review · Reviewer_L249 · 2023-10-27

**Soundness:** 3 good
**Presentation:** 3 good
**Contribution:** 2 fair
**Rating:** 3
**Confidence:** 4

**Summary:**

This paper presents some theoretical results on the convergence and linear stability of SAM, as well as experimental verification.

**Strengths:**

The paper shows that SAM can be flatter than SGD, which is good.

**Weaknesses:**

- First, the theoretical results do not seem to be that new or involved, and the proofs are mostly standard. That can be fine if new insights are uncovered in the paper, but the overall message of the paper is not surprising, and the flatness of SAM solutions is well-studied.

- I also do not understand why overparameterization is stressed so much in the paper, while no result really seems to use overparameterization. I can only see PL condition and overparameterization are discussed, but that is a very hand-wavy discussion. If authors are using a specific result, they should properly refer.

**Questions:**

How does the results of the paper compare to linearization study of https://arxiv.org/pdf/2302.09693.pdf?

---

> ### Author Response · Authors · 2023-11-18
> **Response to L249 (1/2)**
>
> We appreciate the reviewer for taking the time to review our work. While we respond to the reviewer’s specific comments as below, we would be keen to engage in any further discussion.
>
> &nbsp;
>
> **New insights?**
>
> > First, the theoretical results do not seem to be that new or involved, and the proofs are mostly standard. That can be fine if new insights are uncovered in the paper, but the overall message of the paper is not surprising.
>
> We would like to respectfully disagree with the reviewer’s assessment for the following reasons.
>
> First, our theoretical results in Sections 3 and 4 have been neither previously explored nor presented explicitly in prior works. In Section 3, we show that stochastic SAM can achieve a linear convergence rate for overparameterized models, whereas no previous work has focused on the effect of overparameterization, all remaining sublinear convergence rates [1-3]. Notably, this is followed by our experiments showing that these improved convergence properties in theory indeed translate to realistic and practical settings, which has not been shown explicitly in any fashion. In Section 4, we show that SAM finds a flatter solution with bounded eigenvalues of Hessian moments through linear stability analysis. Although the flatness of the SAM solution has been discussed in [10] as pointed out by the reviewer, our results further characterize the eigenvalue divergence of the Hessian moments, which is again substantiated by empirical evaluations.
>
> Also, we would like to stress the significance of our findings in experiments. In Section 6, we discover a strong trend that the generalization benefit of SAM tends to increase with more parameters, and also, that the optimal $\rho$ tends to increase as well as a result of an extensive hyperparameter search. In Section 7, we demonstrate for the first time that sparsity can be a remedy to the computational overhead induced by overparameterization without sacrificing the improved generalization benefit from SAM. We believe that all of these empirical findings can render non-trivial practical values, and can be considered a valuable initial exploration in the literature, as recognized by the reviewer `jLHw` (“*the experiments relating scaling of models to usefulness of SAM are, to my knowledge, novel, and I find them interesting*”) and the reviewer `9tdL` (“*an extensive numerical section verifies the practical utility of the theoretical results*”). Additionally, our results can contribute to new perspectives on SAM's potential in the current landscape of large-scale and efficient training [4, 5].
>
> We sincerely hope that our theoretical results are considered as one of the many contributions in this work, and we provide non-trivial, if not significant, new empirical findings that can provide valuable insights into understanding SAM and render positive avenues for future work.

---

> ### Author Response · Authors · 2023-11-18
> **Response to L249 (2/2)**
>
> **Significance of overparameterization? Any reference?**
>
> > I also do not understand why overparameterization is stressed so much in the paper, while no result really seems to use overparameterization. I can only see PL condition and overparameterization are discussed, but that is a very hand-wavy discussion. If authors are using a specific result, they should properly refer.
>
> Overparameterization does play a crucial role in our theoretical analyses. We elaborate on this as follows.
>
> First off, we characterize overparameterization in Section 2 as a neural network being large enough to interpolate over the whole training data and achieve zero training loss, ensuring the existence of global minima. We formally defined this observation in Definition 1 (interpolation).
>
> This interpolation assumption is prominently used in the proof of linear convergence of stochastic SAM (Theorem 7 of Section 3), which is provided in Appendix D. We utilize the fact obtained from the interpolation assumption that $\nabla f_i (x^\star) = 0$ and $f_i (x^\star) = 0$ for all $i$ in step 3 of the proof of Theorem 12 (linear convergence of mini-batch SAM). To be specific, it modifies the inequality derived from $\beta$-smoothness:
> $$\|\|\nabla f_i(x_t) - \nabla f_i (x^\star)\|\|^2 \leq 2 \beta (f_i(x_t) - f_i(x^\star))$$
> to
> $$\|\|\nabla f_i (x_t)\|\|^2 \leq 2\beta f_i(x_t)$$ and enables us to obtain the linear convergence rate.
>
> Also, we need the interpolation assumption for our linear stability analysis. The linear stability analysis assumes that there exists a fixed point for the optimizer of interest, which is not guaranteed for SGD or stochastic SAM. In our analysis, the interpolation assumption provides the existence of stationary points of the loss function for all individual data points (*i.e.*, $\nabla f_i(x^\star)=0$ for any $i$), which are fixed points of these stochastic optimizers.
>
> We refer to prior works either using the interpolation assumption for linear convergence of SGD [6, 7] or implicitly assuming the existence of a fixed point for linear stability analysis for SGD in [8, 9] in Sections 2, 3, and 4.
>
> &nbsp;
>
> **Comparison to [10]**
>
> >How does the results of the paper compare to linearization study of [10]?
>
> We would like to first acknowledge the reviewer for pointing at [10]; it appears to be very related to our work (Section 4 in particular), and admittedly it is our mistake. We believe it is due to the unfortunate concurrency, but we will certainly update our manuscript to properly cite this work.
>
> First, we find that Section 3 of [10] conducts a linear stability analysis for mini-batch SAM and micro-batch SAM, referred to as SAM and mSAM respectively. As far as we understand, however, this is motivated to show that the proposed micro-batch SAM converges to flatter minima compared to mini-batch SAM.
>
> Clearly, Section 4 of our paper takes a similar approach to analyzing the linear stability of stochastic SAM. However, the main difference lies in revealing that the maximum eigenvalues of the 2nd-4th moments of the stochastic Hessian matrix are bounded. We corroborated these results with empirical evaluations and found that the minima of SAM indeed have more uniformly distributed Hessian moments (see Figure 2c) as predicted in the aforementioned bounds. We would like to note that this is recognized as a strength of our work by the reviewer `jLHw` ("*in particular Equation 7 breaking down the stability condition into necessary conditions on different moments provides a lot of intuition about how SAM might shape network training*").
>
> &nbsp;
>
> **References**
>
> [1] Towards Understanding Sharpness-Aware Minimization, Andriushchenko and Flammarion (2022)\
> [2] Make Sharpness-Aware Minimization Stronger: A Sparsified Perturbation Approach, Mi et al. (2022)\
> [3] Surrogate Gap Minimization Improves Sharpness-Aware Training, Zhuang et al. (2022)\
> [4] Scaling vision transformers to 22 billion parameters, Dehghani et al. (2023)\
> [5] Scaling Laws for Sparsely-Connected Foundation Models, Frantar et al. (2023)\
> [6] The power of interpolation: Understanding the effectiveness of SGD in modern over-parametrized learning, Ma et al. (2018)\
> [7] On exponential convergence of SGD in non-convex over-parametrized learning, Bassily et al. (2018)\
> [8] The alignment property of SGD noise and how it helps select flat minima: A stability analysis, Wu et al. (2022)\
> [9] How SGD selects the global minima in over-parameterized learning: A dynamical stability perspective, Wu et al. (2018)\
> [10] mSAM: Micro-Batch-Averaged Sharpness-Aware Minimization, Behdin et al. (2023)

---

> ### Author Response · Authors · 2023-11-22
> **Last day reminder**
>
> Dear reviewer,
>
> We’d like to gently remind you that the author-reviewer discussion period is ending soon. We’re looking forward to receiving your feedback on our initial response, so we could fix any remaining unclarity and improve our manuscript further. Again, we’re grateful indeed to the reviewer for spending time on our work.
>
> Best regards,\
> Authors

---

> > ### Comment · Reviewer_L249 · 2023-11-22
> >
> > Thank you for your comments.
> >
> > - Use of overparametrization: So if overparametrization is defined as reaching zero loss and interpolation, I don't think this is a very strong result. Smoothness and PL properties are fairly strong assumptions. They are commonly used but I have not seen strong evidence of them holding in practice. Also zero loss is not reached, very often, specially when one uses data augmentation.
> >
> > - Comparison with related work: Thank you for your comparison.
> >
> > Overall,  I don't think the proof techniques are new, and I think the theoretical assumptions are too strong. Therefore, I will keep my evaluation.

---

> ### Author Response · Authors · 2023-11-23
> **Addressing additional comments (1/2)**
>
> Thank you once again for your feedback. We are pleased that our response has assisted in addressing some questions, and we aim to provide further clarifications below.
>
> **Practical example of smoothness and PL properties**
> > Smoothness and PL properties are fairly strong assumptions. They are commonly used but I have not seen strong evidence of them holding in practice.
>
> Although we agree that these assumptions do not apply to all of machine learning, we argue that many common tasks in machine learning and deep learning exhibit smoothness and PL properties. For instance, most large language models possess smooth objectives since smooth functions such as GeLU and Swish are a common choice of activation [1, 2]. In addition, the PL-condition can be shown to hold for most of the parameter space of overparameterized models [3, 4]. Also, in the case of matrix factorization, a widely adopted machine learning algorithm for recommender systems, its non-convex loss exhibits both smoothness and PL properties [5]. We use this to empirically corroborate our convergence result in Figure 1.
>
> Thus, although these assumptions do not always hold for all tasks in machine learning, we would like to argue that they are not too distant from practical settings. We also remark that many works on convergence analyses of various optimization algorithms use these assumptions to gain some insights into its optimization process [5-9].

---

> ### Author Response · Authors · 2023-11-23
> **Addressing additional comments (2/2)**
>
> **Practicality of overparameterization**
> >  Use of overparametrization: So if overparameterization is defined as reaching zero loss and interpolation, I don't think this is a very strong result. … . zero loss is not reached, very often, specially when one uses data augmentation.
>
> First, interpolation can easily hold in the aforementioned matrix factorization by controlling the rank [5]. It can also (nearly) hold for overparameterized neural networks as stated in [10] that “*Modern machine learning paradigms, such as deep learning, occur in or close to the interpolation regime*” or is at least a desideratum in practice as stated in [11] that “*The best way to solve the problem from practical standpoint is you build a very big system ... basically you want to make sure you hit the zero training error*”. We have verified that empirical evaluations under the practical settings in Figures 1 and 2 align well with our theoretical results developed under the interpolation assumption.
>
> Also, we remark that the interpolation assumption has been frequently used in recent literature to show a linear convergence of a stochastic optimizer [12-18] or analyze the linear stability of the minima [19-21]. Specifically starting from the seminal work of [12], many previous works have analyzed the convergence rates under interpolation assumption for various contexts including PL [13], accelerated [14,15], second-order [16], line-search [17], and last-iterate convergence [18]; these works have been appreciated in the community due to their relevances to modern deep learning era. In a similar sense, we believe our analysis in Section 3 can be regarded as one of the many contributions we have made in this work.
>
> Finally, without relying on the interpolation assumption in Sections 5 and 6, our empirical findings highlight the potential of SAM with increasing parameter counts or sparsity levels. We remark that these results are **NOT** empirical verifications of the theorems developed in Sections 3 and 4, and hold significance themselves by showing SAM’s potential in the current landscape of large-scale and efficient learning.
>
> &nbsp;
>
>     In this work, we have analyzed the effects of overparameterization on SAM from both theoretical and empirical perspectives which other reviewers have found to be interesting, novel, and insightful. We would like to believe that we have made quite a reasonable contribution to the community, and we hope that both practitioners and researchers find our findings and discussions helpful.
>
> &nbsp;
>
> **References**\
> [1] A Survey of Transformers, Lin et al. (2022)\
> [2] A Survey of Large Language Models, Zhao et al. (2023)\
> [3] Loss landscapes and optimization in over-parameterized non-linear systems and neural networks, Liu et al. (2022)\
> [4] Fit without fear: remarkable mathematical phenomena of deep learning through the prism of interpolation, Belkin (2021)\
> [5] Stochastic Polyak Step-size for SGD: An Adaptive Learning Rate for Fast Convergence (2021)\
> [6] Towards Understanding Sharpness-Aware Minimization, Andriushchenko and Flammarion (2022)\
> [7] SGD for Structured Nonconvex Functions: Learning Rates, Minibatching and Interpolation (2021)\
> [8] SGDA with shuffling: faster convergence for nonconvex-PŁ minimax optimization, Cho and Yun (2023)\
> [9] Linear Convergence of Adaptive Stochastic Gradient Descent, Xie, Xu, and Ward (2020)\
> [10] Aiming towards the minimizers: fast convergence of SGD for overparametrized problems, Liu et al. (2023)\
> [11] Ruslan Salakhutdinov. Tutorial on deep learning. https://simons.berkeley.edu/talks/ruslan-salakhutdinov-01-26-2017-1 \
> [12] The power of interpolation: Understanding the effectiveness of SGD in modern over-parametrized learning, Ma et al. (2018)\
> [13] On exponential convergence of SGD in non-convex over-parametrized learning, Bassily et al. (2018)\
> [14] Accelerating SGD with momentum for over-parameterized learning, Liu and Belkin (2018)\
> [15] Fast and Faster Convergence of SGD for Over-Parameterized Models (and an Accelerated Perceptron), Vaswani, Bach, and Schmidt (2019)\
> [16] Fast and Furious Convergence: Stochastic Second-Order Methods under Interpolation, Meng et al. (2020)\
> [17] Painless Stochastic Gradient: Interpolation, Line-Search, and Convergence Rates, Vaswani et al. (2019)\
> [18] Last iterate convergence of SGD for Least-Squares in the Interpolation regime, Varre, Pillaud-Vivien, and Flammarion (2021)\
> [19] How SGD selects the global minima in over-parameterized learning: A dynamical stability perspective, Wu et al. (2018)\
> [20] On Linear Stability of SGD and Input-Smoothness of Neural Networks, Ma and Ying. (2021)\
> [21] The alignment property of SGD noise and how it helps select flat minima: A stability analysis, Wu et al. (2022)

---

### Official Review · Reviewer_jLHw · 2023-10-29

**Soundness:** 4 excellent
**Presentation:** 3 good
**Contribution:** 2 fair
**Rating:** 6
**Confidence:** 3

**Summary:**

The authors begin with an analysis of the convergence of SAM, and establish a convergence result for SAM with stochastic gradients by using a smoothness condition which ensures that SAM updates are similar enough to SGD updates for SGD convergence bounds to apply. They then characterize the stability condition for SAM in terms of moments of the stochastic loss Hessian. They conclude with experimental evidence that SAM is more useful for larger models (mixed results for vision transformer), and also is useful for sparsified models.

**Strengths:**

The basic theoretical analysis is clean and easy to follow. In particular Equation 7 breaking down the stability condition into necessary conditions on different moments provides a lot of intuition about how SAM might shape network training. Additionally, the experiments relating scaling of models to usefulness of SAM are, to my knowledge, novel, and I find them interesting. Though the results are more complicated in the case of ViT and ResNet without weight decay, they suggest that this is an area that merits further investigation.

**Weaknesses:**

The basic convergence rate analysis for SAM seems correct but is not very compelling; these types of convergence bounds seem to be far from the rates in practice.

Regarding the experimental results: it is not clear if the effects are due to the networks pushing into the interpolation regime. For example in the MNIST and CIFAR examples, the number of parameters is much larger than the number of datapoints for most of the examples, but the gap does not develop until well into this regime. I have listed some questions about this phenomenology below; I believe some more detail on this point could make the paper significantly stronger.

Update: After significant engagment by the authors in the review period, some of the weaknesses have been addressed, and I updated my review score.

**Questions:**

It would be helpful to define $f_i$ and its relationship to $f$ more explicitly.

How does batch size play a role in the analysis and the various theorems?

The paper claims that as models become more overparameterized, SAM becomes more useful. What is the evidence that the models in the experiments are distinct in their level of overparameterization? What fraction of them are reaching interpolating minima? How close are any of the settings to the NTK regime?

One interesting experiment could be to train a networks with a fixed number of parameters, but which is closer to or further from the linearized regime (using techniques from [1] and [2]), with and without SAM, and seeing if SAM is more helpful in the linearized regime or not. This could provide another insight on whether or not overparameterization itself is the cause for the differences in effectiveness.

[1] https://proceedings.neurips.cc/paper_files/paper/2019/hash/ae614c557843b1df326cb29c57225459-Abstract.html
[2] https://arxiv.org/abs/2010.07344

---

> ### Author Response · Authors · 2023-11-18
> **Response to jLHw (1/2)**
>
> We are really encouraged by the reviewer’s positive and constructive feedback. We believe this has led us to improve our work quite significantly. While we respond to the reviewer’s specific comments as below, we would be keen to engage in any further discussion.
>
> &nbsp;
>
> **On the significance of convergence result**
> > The basic convergence rate analysis for SAM seems correct but is not very compelling; these types of convergence bounds seem to be far from the rates in practice.
>
> We would like to respectfully argue that our convergence result is non-trivial, if not significant, for the following reasons. While the convergence properties of stochastic SAM have been studied recently [1-3], they all show a *sublinear* convergence rate. Alternatively, we show that SAM can achieve a *linear* convergence rate, which is much faster than a sublinear one. This level-shifting improvement in theory is also confirmed by our experiments as shown in Figure 1, in which we show that SAM, for overparameterized models, can indeed accelerate in practical settings, which is referred to as “*an extensive numerical section verifies the practical utility of the theoretical results*” by the reviewer `9tdL`.
>
> &nbsp;
>
> **On overparameterization in experiments**
>
> > Regarding the experimental results: it is not clear if the effects are due to the networks pushing into the interpolation regime. For example in the MNIST and CIFAR examples, the number of parameters is much larger than the number of datapoints for most of the examples, but the gap does not develop until well into this regime.
> > What is the evidence that the models in the experiments are distinct in their level of overparameterization? What fraction of them are reaching interpolating minima?
>
> We clarify first that the number of parameters being larger than that of data points does not equate to overparameterization. In our experiments, overparameterization simply refers to the act of increasing the number of parameters, and as a result, we show that the generalization improvement by SAM tends to increase. We suspect that the confusion perhaps originates from our usage of the term overparameterization, to interchangeably refer to interpolation in theory in Sections 3 and 4. We admit that this can easily confuse the readers, and thus, will certainly make it clearer in the revised version. Nevertheless, we supplement plots for training loss plots vs. # of parameters in Figure 9 in Appendix B to show where a model begins to interpolate, *i.e.*, to reach (almost) zero training loss.
>
> &nbsp;
>
> **SAM in linearized regime**
>
> > One interesting experiment could be to train a networks with a fixed number of parameters, but which is closer to or further from the linearized regime with and without SAM, and seeing if SAM is more helpful in the linearized regime or not. This could provide another insight on whether or not overparameterization itself is the cause for the differences in effectiveness.
>
> |     | $\alpha=1$ |  | $\alpha=1000$  |
> |-----|--------|---|---|
> |     | acc $\hspace{1.5em}$ stability |$\hspace{1.5em}$| acc $\hspace{1.5em}$ stability  |
> | SGD | $88.33 \hspace{1em} 0.53$  || $53.11 \hspace{1em}0.99$  |
> | SAM( $\rho = 0.001$)| $87.95 \hspace{1em} 0.53$ || $40.71  \hspace{1em} 0.99$ |
> | SAM ($\rho = 0.01$)  | $88.48 \hspace{1em} 0.53$  || $11.42 \hspace{1em}0.90$  |
>
> To evaluate the effect of linearization on SAM (and differentiate it from that of overparameterization), we conduct experiments on VGG-11/Cifar-10 following the same setting of [8]. The results are presented in the table above; here, ‘acc’ and ‘stability’ each represent test accuracy and stability of activations; $\alpha$ is a parameter to control how close the network is to the linearized regime, and as a result of a larger $\alpha$, the stability of activations being close to 1 corresponds to a large degree of linearization.
>
> We first find that, in the highly linearized regime, SAM can even underperform SGD.
> Precisely, SGD and SAM (with $\rho = 0.001$) both achieve the same level of effective linearization at $\alpha = 1000$, *i.e.*, where stability is near $1.0$, and yet, SAM is outperformed by SGD by more than $10\%$.
>
> This means that the benefit of SAM for improving generalization in our experiments is not attributed to linearization; rather, it is due to the increased number of parameters we controlled.
>
> We have included these findings and discussion in more detail in Appendix C of the revised paper. Once again, we appreciate the reviewer for promoting fruitful discussion.

---

> > ### Author Response · Authors · 2023-11-18
> > **Response to jLHw (2/2)**
> >
> > **Role of batch size**
> >
> > > How does batch size play a role in the analysis and the various theorems?
> >
> > We extend Theorem 7 to Theorem 12 in Appendix D as a mini-batch version. This result shows that increasing the batch size $B$ leads to a larger step size $\eta^\star$ and makes SAM converge with fewer steps, which is consistent with existing work [4-6]. This speedup, however, does not scale linearly with increasing $B$ and will be saturated after some point called critical batch size. We remark that our analysis can be used to estimate a critical batch size similarly to [4], and further to its relationship with $\rho$. We leave this for future work.
> >
> > Theorem 9 (and Equation 7) can be extended to involve the role of batch size as well. This shows that the upper bounds on sharpness and Hessian non-uniformity decrease with the decreasing batch size, which is similar to the case of SGD [7].
> >
> > &nbsp;
> >
> > **Relationship between $f_i$ and $f$**
> >
> > > It would be helpful to define $f_i$ and its relationship to $f$ more explicitly.
> >
> > $f_i$ is related to $f$ by $f(x) = \sum_{i=1}^n f_i(x)$ where $n$ is the total number of data points. We will make this more explicit in the revised paper.
> >
> > &nbsp;
> >
> > **References**\
> > [1] Towards Understanding Sharpness-Aware Minimization, Andriushchenko and Flammarion (2022)\
> > [2] Make Sharpness-Aware Minimization Stronger: A Sparsified Perturbation Approach, Mi et al. (2022)\
> > [3] Surrogate Gap Minimization Improves Sharpness-Aware Training, Zhuang et al. (2022)\
> > [4] The power of interpolation: Understanding the effectiveness of SGD in modern over-parametrized learning, Ma et al. (2018)\
> > [5] Don't Decay the Learning Rate, Increase the Batch Size, Smith et al. (2018)\
> > [6] A Bayesian Perspective on Generalization and Stochastic Gradient Descent, Smith and V. Le (2018)\
> > [7] How SGD selects the global minima in over-parameterized learning: A dynamical stability perspective, Wu et al. (2018)\
> > [8] On Lazy Training in Differentiable Programming, Chizat et al. (2019)

---

> > > ### Comment · Reviewer_jLHw · 2023-11-20
> > > **Response to author revisions**
> > >
> > > I thank the authors for their detailed response; some followup comments below.
> > >
> > > Regarding the definition of overparameterization: I understand the setting better now, thanks for the clarification. In this case I do think it is important for Figures 3 and 4 to show information about the training loss/accuracy so the reader can better map parameter count to the true quantity one cares about (interpolation). Can the authors upload revised figures which show this information?
> > >
> > > I also thank the reviewers for the experiments in Section C. Can you confirm that the training loss goes to 0 in the linearized case, at least for small values of $\rho$? The results of this appendix should also be referenced in the main text, as I think they support the interpretation that interpolation is the main feature of settings where SAM performs well.

---

> ### Author Response · Authors · 2023-11-21
> **Addressing additional comments**
>
> We sincerely thank the reviewer for the response. We are glad that some confusion has been clarified. We address the additional comments below.
>
> &nbsp;
>
> **Plot of interpolation vs. parameter count**
>
> >In this case I do think it is important for Figures 3 and 4 to show information about the training loss/accuracy so the reader can better map parameter count to the true quantity one cares about (interpolation). Can the authors upload revised figures which show this information?
>
> We have updated the paper to include the full training loss plots in Figures 3 and 4. We are really appreciative to the reviewer for this suggestion. This will certainly promote a better understanding.
>
> &nbsp;
>
> **Training loss in linearized regime**
>
> >Can you confirm that the training loss goes to 0 in the linearized case, at least for small values of $\rho$? The results of this appendix should also be referenced in the main text, as I think they support the interpretation that interpolation is the main feature of settings where SAM performs well.
>
> |     | $\alpha=1$ | $\alpha=1000$  |
> |-----|--------|---|
> | SGD | $0.002$ |$0.069$  |
> | SAM( $\rho = 0.001$)| $0.002$ |$0.087$ |
>
> We have measured the training losses for both SGD and SAM, and the results are reported in the table above. While both SGD and SAM achieve a low (close-to-zero) training loss of $0.002$ at $\alpha=1$, their training losses both increase at $\alpha=1000$, *i.e.*, the training loss (for neither SGD nor SAM) does not go to $0$ in the linearized regime.
>
> Importantly, please note that this is expected and consistent with [1], which shows that the linear model obtained with large $\alpha$ could not reach high training accuracies despite overparameterization; see Figure 3(a) for decreasing training accuracy and Figure 7(a) for increasing training loss in [1] as $\alpha$ increases. This means that, once again, the benefit of SAM is *NOT* due to linearization (or interpolation), but is due to overparameterization.
>
> Nonetheless, we have mentioned this in Section 5 of the main text, and also included the full discussion in Appendix C.
>
> &nbsp;
>
>     We hope that our response has adequately addressed your comments, but please let us know if there is anything else you want us to address further. We will make our best efforts to reflect that as well. After all, we would like to believe that we have made a reasonable contribution to the community. In this respect, we would greatly appreciate it if the reviewer could give a re-consideration to the initial rating of this work.
>
> &nbsp;
>
>
> **References**\
> [1] On Lazy Training in Differentiable Programming, Chizat et al. (2019)

---

> > ### Comment · Reviewer_jLHw · 2023-11-21
> > **Thanks for the responses**
> >
> > I appreciate the thoroughness of the responses. I have updated my review score to take into account the updates.

---

> > > ### Author Response · Authors · 2023-11-22
> > > **Thank you**
> > >
> > > We genuinely appreciate your decision to raise the score and provide constructive feedback to improve the paper. We will make sure to incorporate your suggestions into the revised version.

---

### Official Review · Reviewer_4Lps · 2023-10-31

**Soundness:** 3 good
**Presentation:** 3 good
**Contribution:** 3 good
**Rating:** 6
**Confidence:** 3

**Summary:**

In this paper, they empirically and theoretically show the effect of overparameterization on SAM. By defining interpolation in terms of overparameterization, they demonstrate that SAM converges at a linear rate under such conditions. Additionally, they illustrate that SAM can achieve flatter minima than SGD. Varying the number of parameters, they empirically show the effect of overparameterization.

**Strengths:**

This paper presents mathematical theories to verify the impact of overparameterization on SAM and the relationship between SGD and SAM.

**Weaknesses:**

Minor weaknesses in the paper include typos, as observed in Figure 4 where image captions and labels do not match (e.g., (a,b) → (a), (c,d) → (b)). Additionally, to enhance clarity, it's advisable to consider indexing terms more carefully. For instance, changing 's_i for i-th moment' to 's_k for k-th moment' on page 4 may help avoid confusion.

**Questions:**

I think the assumptions to be rather stringent: for example, requiring $\beta$-smoothness for each individual point. Is that assumption based on the model's overparameterization? Is there any justification for the assumption?
As I understand it, as the number of parameters increases, the model needs to be smoother, which should lead to an increase in the optimal perturbation bound. However, in Figure 5(c), the optimal rho decreases when the number of parameters is 14.8m. Is there any explanation for this phenomenon?

---

> ### Author Response · Authors · 2023-11-18
> **Response to 4Lps**
>
> We really appreciate the reviewer’s positive and constructive feedback. We believe this has led us to improve our work to a large extent. While we respond to the reviewer’s specific comments as below, please do let us know if there is anything else we can address further.
>
> &nbsp;
>
> **Individual $\beta$-smoothness assumption**
> > I think the assumptions to be rather stringent: for example, requiring $\beta$-smoothness for each individual point. Is that assumption based on the model's overparameterization? Is there any justification for the assumption?
>
> We would like to note first that the assumption on the $\beta$-smoothness for each $f_i$ is quite standard and frequently used in the optimization literature [1-5], which is also referred to as ``standard smoothness’’ by the reviewer `9tdL`. This assumption is considered fairly mild in that it is satisfied for neural networks with smooth activation and loss function with bounded inputs, for example, cross-entropy loss for normalized images. This assumption has little to do with overparameterization.
>
> &nbsp;
>
> **Non-monotonicity in Figure 5(c)**
> >However, in Figure 5(c), the optimal rho decreases when the number of parameters is 14.8m. Is there any explanation for this phenomenon?
>
> We first fixed the typo in the x-axis of Figure 5 (c) in the revised version: *i.e.*, from [...14.8m 14.8m 25.6m] to [... 14.8m 25.6m 39.3m]. Now regarding the non-monotonic spot, *i.e.*, $\rho^\star=0.02$ at 25.6m, we believe that it is within a margin of statistical error given that the number of samples evaluated was only 3. Specifically, in a more statistically robust result we provided in the paper, *i.e.*, in Figure 13 in Appendix B, it appears that $\rho^\star$ tends to increase with larger models. We also note that the accuracies at $\rho^\star = 0.02$ and at the second best $\rho = 0.05$ are close to each other with the difference being only $0.04\%$. Furthermore, SAM with $\rho=0.1$ underperforms SGD at 14.8m, while it outperforms SGD at 25.6m model. Based on this evidence, we believe that this result can be potentially refined to show a more monotonic trend under an increased experiment budget.
>
> &nbsp;
>
> **(additionally) On the optimal perturbation bound**
> > As I understand it, as the number of parameters increases, the model needs to be smoother, which should lead to an increase in the optimal perturbation bound.
>
> We believe that your understanding aligns well with ours: smoother models can lead to an increase of the optimal perturbation bound in the following sense.
>
> First, it is well known that the optimal step size for standard gradient methods is given by $1/\beta$ (where $\beta$ denotes the bound on the smoothness of $f$) [6, 7]. Since the perturbation bound of SAM corresponds to the step size used in the inner maximization step of SAM (*i.e.*, the perturbation step), this means that the optimal perturbation bound $\rho^\star$ should increase as the model becomes smoother with less $\beta$.
>
> We can also relate $\beta$ and $\rho$ via the perspective of securing enough effect of SAM. More precisely, the Lipschitzness of SAM’s one-step update can be expressed as follows:
> $$\Bigg\lVert \nabla f \left(x+ \rho \frac{\nabla f(x)}{\lVert \nabla f(x) \rVert} \right) - \nabla f(x) \Bigg\rVert \leq \beta \left\lVert x+ \rho\frac{\nabla f(x)}{\lVert \nabla f(x) \rVert}- x \right\rVert = \beta \rho.$$
> Notice that with a smaller $\beta$ (*i.e.*, a smoother model), a larger $\rho$ is required to maintain the same level of Lipschitzness of the SAM gradient. This means that in order to enjoy the potential benefit of SAM, the optimal perturbation bound needs to increase for a smoother $f$, provided by overparameterization.
>
> We believe that this discussion helps us to reinforce our understanding of the behavior of the optimal perturbation bound further, and thus, will certainly include it in the revised version.
>
> &nbsp;
>
> **Typos**\
> We have fixed them in the revised version following your suggestions.
>
> &nbsp;
>
> **References**\
> [1] Accelerating Stochastic Gradient Descent using Predictive Variance Reduction, Johnson and Zhang (2013)\
> [2] SCAFFOLD: Stochastic Controlled Averaging for Federated Learning, Karimireddy et al. (2020)\
> [3] Optimal Rates for Random Order Online Optimization, Sherman et al. (2021)\
> [4] Lower Complexity Bounds for Finite-Sum Convex-Concave Minimax Optimization Problems, Xie et al. (2020)\
> [5] Towards Understanding Sharpness-Aware Minimization, Andriushchenko and Flammarion (2022)\
> [6] Convex Optimization, Boyd and Vandenberghe (2004)\
> [7] Convex Optimization: Algorithms and Complexity, Bubeck (2015)

---

> > ### Author Response · Authors · 2023-11-22
> > **Last day reminder**
> >
> > Dear reviewer,
> >
> > Just a friendly reminder that the deadline for author-reviewer discussions period is closing soon. We've tried to address the reviewer's questions in our recent respose and remain open to any additional questions or suggestions. We again extend our gratitude to the reviewer for their dedicated review and valuable feedback to help polish our work.
> >
> > Sincerely,\
> > Authors

---

> > > ### Comment · Reviewer_4Lps · 2023-11-23
> > >
> > > Thank you for your answering. I will keep my positive rating.

---

### Official Review · Reviewer_9tdL · 2023-11-01

**Soundness:** 3 good
**Presentation:** 3 good
**Contribution:** 3 good
**Rating:** 8
**Confidence:** 3

**Summary:**

This paper studies sharpness-aware minimization under differing levels of overparameterization. The authors obtain linear convergence and stability of the obtained minima in the interpolating regime, under standard smoothness and PL conditions. An extensive numerical section verifies the practical utility of the theoretical results, and further investigates the effects of sparsification as a method to alleviate computational burden.

**Strengths:**

This appears to be a solid paper, recovering versions of results for SGD for a relevant related problem. The paper is generally well written and presented.

**Weaknesses:**

Further discussion on the non-monotonicity encountered in the experimental section would be useful. Reference to where and when the model starts to interpolate the data, as well as how the peaks in relative accuracy and accuracy correspond to these or others points the authors may have observed to be relevant, would be interesting.

**Questions:**

-

**Details Of Ethics Concerns:**

-

---

> ### Author Response · Authors · 2023-11-18
> **Response to 9tdL**
>
> We really appreciate the reviewer’s positive and constructive feedback. While we address the reviewer’s specific comments below, we would be keen to engage in any further discussion.
>
> &nbsp;
>
> **Non-monotonicity in experiments and its relation to interpolation**
>
> > Further discussion on the non-monotonicity encountered in the experimental section would be useful. Reference to where and when the model starts to interpolate the data, as well as how the peaks in relative accuracy and accuracy correspond to these or others points the authors may have observed to be relevant, would be interesting.
>
>
> We first report where the models interpolate data in Appendix B; please see Figures 9 and 10 for the training losses of the models used in Figures 3 and 4 in Section 5, respectively. We note that the progress of interpolation with an increasing degree of parameterization is perhaps best viewed in the cases of Cifar-10.
>
> Now, while we categorize the non-monotonicity spotted in Figure 3 as a margin of error given that the overall increasing trend tends to remain, we find that the non-monotonic behaviors observed in Figure 4 are certainly not coincidental. In terms of whether it is related to interpolation locations, here is our observation in summary (again, see Figures 9 and 10 for more details):
>
> |                 | rel acc peak | interpolation | abs acc peak |
> |-----------------|------------------------|---------------------|---------------|
> | ViT             | 209k                   | 3m                  | 3m            |
> | ResNet (w/o wd) | 176k                   | 3m                  | 11m           |
>
> *i.e.*, with increasing degree of parameterization, interpolation is located in the following order: relative accuracy peak < interpolation $\leq$ absolute accuracy peak.
>
> We suspect that the benign effect of overparameterization (and SAM) is not always guaranteed. In particular, it appears that in cases where it is prone to overfitting, this effect can diminish. Specifically, despite the fact that the Cifar-10 experiments share every configuration, except that Figure 4 is obtained either without weight decay (*i.e.*, no regularization) or with the different architecture of ViT (*i.e.*, limited inductive bias compared to convolutional architecture), we see only in Figure 4 that neither absolute accuracy nor relative accuracy gain by SAM increases as with the increasing degree of overparameterization.
>
> We further conjecture that the range of parameterization for which the positive effect in generalization is obtained can either expand or reduce depending on some implicit (*e.g.*, inductive bias) and/or external factors (*e.g.*, regularization) that are unfortunately not precisely comprehended as of yet. Nonetheless, it is certainly a very interesting phenomenon, and we will include this discussion in more detail in the revised version. We would like to thank the reviewer again for appreciating our work.

---

> > ### Comment · Reviewer_9tdL · 2023-11-22
> >
> > Thanks for your response. The inclusion of the mentioned discussion will be useful, and I have not changed my (positive) score.

---

> > > ### Author Response · Authors · 2023-11-22
> > > **Thank you**
> > >
> > > Once again, we express our sincere gratitude for your positive review and valuable feedback to further enhance our paper. We will make our best efforts to reflect your suggestions in the final version.

---

### Public Comment · ~Kyunghun_Nam1 · 2023-11-23
**Questions regarding your paper [1]**

Dear,
I read your paper with great interest.\
ICLR aims to foster active participation within the community by openly sharing all processes from paper submission to the final decision via OpenReview, almost in real-time.
Therefore, despite the ongoing paper review process, I want to leave comments discussing my thoughts and impressions after reading this paper.
Firstly, I thoroughly read the main text and briefly skimmed the proofs in the Appendix.

As far as I comprehend, it seems the authors are defining the finite-sum problem. \
That is, authors derived an iteration complexity upper bound for the SAM algorithm in the problem of $f(x) = \sum_{i=1}^n f_i(x)$.\
Moreover, they demonstrated that when the model is overparameterized, SAM in its stochastic version exhibits a linear rate under properly tuned step size ($\eta$) and perturbation size  ($\rho$).

I'd like to delve into the discussion regarding the overparameterized setting first. \
Definition 1 elucidates overparameterzation by suggesting that $f(x)$ can converge to $0$, and at the optimal parameter $x^{\star}$, the sample loss $f_i(x^{\star})$ and the sample gradient $\nabla f_i(x^{\star})$ become $0$. I am well aware that this is a quite plausible scenario in an overparameterization situation.

However, I am not entirely certain about the difference between your Definition 1 and the following assumptions:
1. $f(x)$ is lower bounded and its infimum is $0$, $i.e.,$ $f(x) \ge f^{\inf} = 0$.
2. $\mathbb{E}[\lVert \nabla f_i(x) \rVert^2] \le \sigma_0 + \sigma_1 \lVert \nabla f(x) \rVert^2$.
for some $\sigma_0, \sigma_1 \ge 0$.

In the optimization field, Assumption $2$ refers to the 'Affine Variance (second moment) assumption' \
Under the "strong growth setting" ($\sigma_0 = 0$), if $\lVert \nabla f(x) \rVert = 0$, then $\lVert \nabla f_i(x) \rVert =0$ for all $i \in [n]$.
This is an assumption that includes overparameterized settings.

Hence, instead of solely focusing on the overparameterization setting, I believe it would be more meaningful to demonstrate SAM's iteration complexity bound under Assumption 1 and Assumption 2. (Theorem)\
Subsequently, showing through corollary that when $\sigma_0 = 0$, SAM exhibits a linear convergence rate.

Next, I think this paper should necessarily reference the following paper.\
"Practical Sharpness-Aware Minimization Cannot Converge All the Way to Optima" Si et al.
(Neurips 2023)

In that paper, by establishing the iteration complexity "lower" and upper bounds for the SAM algorithm, it was shown that SAM converges to a bounded region (near stationary points) and the size of this region depends on SAM's hyperparameters. ($e.g., \eta, \rho$)
I consider this paper quite meaningful. However, since you've shown that SAM not only converges to stationary points but also converges with a linear convergence rate, it's necessary to explain the difference from the aforementioned paper.

Possibly, this distinction could be linked back to assumption 2. (affine variance assumption)
When $\sigma_0 \neq 0$, SAM may not converge to a stationary point. However, in a scenario where $\sigma_0 = 0$, such as in an overparameterized, claiming convergence to a stationary point might have been possible.

---

> ### Public Comment · ~Kyunghun_Nam1 · 2023-11-23
> **Questions regarding your paper[2]**
>
> Next, I'm curious about the practicality of $\eta$ and $\rho$ in your theorem.\
> While the paper focuses on SAM's iteration complexity bound in an overparameterized scenario, which is relatively common in today's large-scale ML/DL, the conditions for $\rho$ and $\eta$ are theorem seem impractical.\
> I believe it should be for any $\rho$ (at least).  This is because $\rho$ is an added hyperparameter of SAM that SGD does not have.
> The $\eta$ and $\rho$ in the theorem require prior information (prior knowledge) about the optimization problem and objective function ($e.g., \beta, \alpha$) etc.\
> Notably, SGD is known to achieve a (nearly) optimal convergence rate when the step size is appropriately tuned using prior knowledge. Therefore, it remains ambiguous whether your theoretical results (linear convergence rate) truly present a SAM-specific advantage (over SGD) in overparameterization scenarios or are a consequence of finely tuned-hyperparameters.
>
> Lastly, I'm curious why in Appendix, Lemma 10 and 11 and thereafter $x_{t+ \frac{1}{2}} = x_t + \rho \nabla f_i(x_t) $ was chosen. \
> This deviates from SAM's original update rule. The gradient normalization term (in the ascent step) holds significant importance.
> For further insights into this matter, I would recommend referencing the paper titled,\
> "The Crucial Role of Normalization in Sharpness-Aware Minimization" Dai et al. (Neurips 2023)"
>
> Finally, two minor questions remain.\
> PL-function can be thought of as "a nonconvex generalization of strongly convex functions".  This assumption appears unrelated to overparameterization.\
>  In other words, this property has nothing to do with overparameterized, so I don't understand why this assumption was used. \
> Moreover, it is well known that under the PL-condition, the first-order algorithm has a linear convergence rate.\
> Next, why did you assume $\beta$-smoothness of $f_i$ and $\lambda$-smoothness of $f$ at the same time? If assume $\beta$-smoothness of $f_i$ in the finite-sum problem, then the smoothness of $f$ can also be expressed in terms of $\beta$, i.e., there is no need to introduce $\lambda$.
>
> I'd like to talk about my feelings about the second (main) contribution of this paper, but I realize that would make this post too long, so I'll cut it short.\
> Thank you for considering these points. I look forward to any elucidation or discussions regarding these opinions.
>
> Best regards
>
> Reference :
> 1. RMSProp converges with proper hyper-parameters. Shi et al. (ICLR 2021)
> 2. Adam can converge without any modification on its update rules. Zhang et al. (Neurips 2023)
> 3. The power of Adaptivity in SGD: Self-tuning step sizes with Unbounded gradients and Affine variance. Faw et al. (COLT 2022)
> 4, Lower bounds for finding stationary points 1. Carmon et al. (arXiv 1710.11606)
> 5. Handbook of Convergence theorems for (Stochastic) Gradient Methods. Gower et al.

---

### Author Response · Authors · 2023-11-23
**Thanks to the reviewers**

Dear reviewers,

As the end of discussion period is closing, we’d like to sincerely thank all the reviewers for spending their time and efforts on reviewing our paper and providing constructive feedbacks, from simple typos, to elaborating on the interpretation of certain experiments, suggesting new and interesting experiment and discussions on the details of the theories. During this period, we put our best efforts to address reviewers’ comments and provide clarifications. Also, we believe that through insightful conversations and discussions with the reviewers, we refined our understandings on different subjects which contributed to further enhance our paper. Finally, we promise to incorporate suggestions that has yet to be included in the revised version.

Sincerely,\
Authors

---

### Meta-Review · Area_Chair_yMyf · 2023-12-16

**Metareview:**

This work investigated the effect of overparameterization on SAM both empirically and theoretically. By defining interpolation in terms of overparameterization, the work demonstrated that SAM converges at a linear rate. Additionally, the work justifies that SAM can achieve flatter minima than SGD.

Although most reviewers appreciate the empirical findings, the major concern of the work is on the theoretical results.
The PL condition cannot be justified on practical deep networks, and the theoretical results are incremental compared to existing results.

**Justification For Why Not Higher Score:**

the theoretical study is incremental with strong assumptions that might not hold in practice

**Justification For Why Not Lower Score:**

n/a

---

### Decision · Program_Chairs · 2024-01-16

Reject